# A General Framework for Equivariant Neural Networks on Reductive Lie Groups

**Ilyes Batatia**
Engineering Laboratory,
University of Cambridge
Cambridge, CB2 1PZ UK
Department of Chemistry,
ENS Paris-Saclay, Université Paris-Saclay
91190 Gif-sur-Yvette, France
`ilyes.batatia@ens-paris-saclay.fr`

**Mario Geiger**
Department of Electrical Engineering
and Computer Science,
Massachusetts Institute of Technology
Cambridge, MA, USA

**Jose Munoz**
EIA University, FTA Group
Antioquia, Colombia

**Tess Smidt**
Department of Electrical Engineering
and Computer Science,
Massachusetts Institute of Technology
Cambridge, MA, USA

**Lior Silberman**
Department of Mathematics
University of British Columbia
Vancouver, BC, Canada V6T 1Z2

**Christoph Ortner**
Department of Mathematics
University of British Columbia
Vancouver, BC, Canada V6T 1Z2

## Abstract

Reductive Lie Groups, such as the orthogonal groups, the Lorentz group, or the unitary groups, play essential roles across scientific fields as diverse as high energy physics, quantum mechanics, quantum chromodynamics, molecular dynamics, computer vision, and imaging. In this paper, we present a general Equivariant Neural Network architecture capable of respecting the symmetries of the finite-dimensional representations of any reductive Lie Group $G$. Our approach generalizes the successful ACE and MACE architectures for atomistic point clouds to any data equivariant to a reductive Lie group action. We also introduce the `lie-nn` software library, which provides all the necessary tools to develop and implement such general $G$-equivariant neural networks. It implements routines for the reduction of generic tensor products of representations into irreducible representations, making it easy to apply our architecture to a wide range of problems and groups. The generality and performance of our approach are demonstrated by applying it to the tasks of top quark decay tagging (Lorentz group) and shape recognition (orthogonal group).

## 1 Introduction

Convolutional Neural Networks (CNNs) (LeCun *et al.*, 1989) have become a widely used and powerful tool for computer vision tasks, in large part due to their ability to achieve translation equivariance. This property led to improved generalization and a significant reduction in the number of parameters. Translation equivariance is one of many possible symmetries occurring in machine learning tasks.

37th Conference on Neural Information Processing Systems (NeurIPS 2023).

A wide range of symmetries described by reductive Lie Groups is present in physics, such as $O(3)$ in molecular mechanics, $SO(1,3)$ in High-Energy Physics, $SU(2^N)$ in quantum mechanics, and $SU(3)$ in quantum chromodynamics. Machine learning architectures that respect these symmetries often lead to significantly improved predictions while requiring far less training data. This has been demonstrated in many applications including 2D imaging with $O(2)$ symmetry (Cohen and Welling, 2016a; Esteves *et al.*, 2017), machine learning force fields with $O(3)$ symmetry (Anderson *et al.*, 2019; Bartók *et al.*, 2013; Batzner *et al.*, 2022; Batatia *et al.*, 2022a) or jet tagging with $SO^+(1,3)$ symmetry (Bogatskiy *et al.*, 2022; Li *et al.*, 2022).

One way to extend CNNs to other groups (Finzi *et al.*, 2020; Kondor and Trivedi, 2018) is through harmonic analysis on homogeneous spaces, where the convolution becomes an integral over the group. Other architectures work directly with finite-dimensional representations. We follow the demonstration of Bogatskiy *et al.* (2020a) who constructed a universal approximation of any equivariant map with a feed-forward neural network with vector activations belonging to finite-dimensional representations of a wide class of Lie groups. In this way, one can avoid computational challenges created by infinite-dimensional representations.

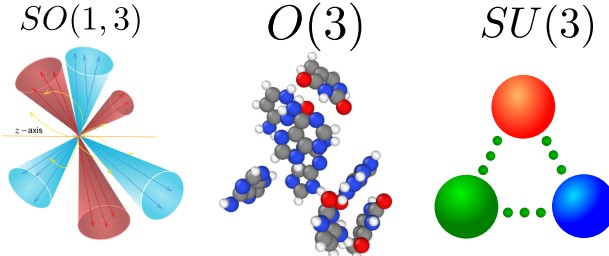

Figure 1: Examples of natural science problems and associated reductive Lie groups. For high energy physics, the Lorentz group $SO(1,3)$; for chemistry, the Euclidean group $E(3)$; for quantum-chromodynamics, the $SU(3)$ group.

Alternatively, our current work can be thought of as a generalization of the Atomic Cluster Expansion (ACE) formalism of Drautz (2019) to general Lie groups. The ACE formalism provides a complete body-ordered basis of $O(3)$-invariant features. By combining the concepts of ACE and $E(3)$-equivariant neural networks, Batatia *et al.* (2022a) proposed the MACE architecture, which achieves state-of-the-art performance on learning tasks in molecular modelling. The present work generalizes the ACE and MACE architectures to arbitrary Lie groups in order to propose a generic architecture for creating representations of geometric point clouds in interaction.

Concretely, our work makes the following contributions:

- We develop the $G$-Equivariant Cluster Expansion. This framework generalizes the ACE (Drautz, 2019) and MACE (Batatia *et al.*, 2022b) architectures to parameterize properties of point clouds, equivariant under a general reductive Lie group $G$.
- We prove that our architecture is universal, even for a single layer.
- We introduce `lie-nn`, a new library providing all the essential tools to apply our framework to a variety of essential Lie Groups in physics and computer visions, including the Lorentz group, $SU(N)$, $SL_2(\mathbb{C})$ and product groups.
- We illustrate the generality and efficiency of our general-purpose approach by demonstrating excellent accuracy on two prototype applications, jet tagging, and 3D point cloud recognition.

## 2   Background

We briefly review a few important group-theoretic concepts: A real (complex) **Lie group** is a group that is also a finite-dimensional smooth (complex) manifold in which the product and inversion of the group are also smooth (holomorphic) maps. Among the most important Lie groups are Matrix Lie groups, which are closed subgroups of $GL(n, \mathbb{C})$ the group of invertible $n \times n$ matrices with complex entries. This includes well-known groups such as $Sp(2n, \mathbb{R})$ consisting of matrices of determinant one, that is relevant in Hamiltonian dynamics A finite-dimensional **representation** of the Lie group

$G$ is a finite-dimensional vector space $V$ endowed with a smooth homomorphism $\rho\colon G \to \mathrm{GL}(V)$. Features in the equivariant neural networks live in these vector spaces. An **irreducible** representation $V$ is a representation that has no subspaces which are invariant under the action of the group (other than $\{0\}$ and $V$ itself). This means that $V$ can not be decomposed non-trivially as the direct sum of representations. A **reductive group** over a field $F$ is a (Zariski-) closed subgroup of the group of matrices $\mathrm{GL}(n, F)$ such that every finite-dimensional representation of $G$ on an $F$-vectorspace can be decomposed as a sum of irreducible representations.

## 3  Related Work

**Lie group convolutions**  Convolutional neural networks (CNNs), which are translation equivariant, have also been generalized to other symmetries. For example, G-convolutions (Cohen and Welling, 2016b) generalized CNNs to discrete groups. Steerable CNNs (Cohen and Welling, 2016a) generalized CNNs to $O(2)$ equivariance and Spherical CNNs (Cohen *et al.*, 2018) $O(3)$ equivariance. A general theory of convolution on any compact group and symmetric space was given by Kondor and Trivedi (2018). This work was further extended to equivariant convolutions on Riemannian manifolds by Weiler *et al.* (2021).

**ACE**  The Atomic Cluster Expansion (ACE) (Drautz, 2019) introduced a systematic framework for constructing complete $O(3)$-invariant high body order basis sets with constant cost per basis function, independent of body order (Dusson *et al.*, 2022).

**e3nn + Equivariant MLPs**  The e3nn library (Geiger and Smidt, 2022) provides a complete solution to build $E(3)-$equivariant neural networks based on irreducible representations. The Equivariant MLPs (Finzi *et al.*, 2021) include more groups, such as $SO(1,3)$, $Z_n$, but are restricted to reducible representations making them much less computationally efficient than irreducible representations.

**Equivariant MPNNs and MACE**  Equivariant MPNNs (Kondor *et al.*, 2018; Anderson *et al.*, 2019; Bogatskiy *et al.*, 2020a; Satorras *et al.*, 2021; Brandstetter *et al.*, 2022; Batzner *et al.*, 2022) have emerged as a powerful architecture to learn on geometric point clouds. They construct permutation invariants and group equivariant representations of point clouds. Successful applications include simulations in chemistry, particle physics, and 3D vision. MACE (Batatia *et al.*, 2022a) generalized the $O(3)$-Equivariant MPNNs to build messages of arbitrary body order, outperforming other approaches on molecular tasks. (Batatia *et al.*, 2022b) showed that the MACE design space is large enough to include most of the previously published equivariant architectures.

## 4  The $G$-Equivariant Cluster Expansion

We are concerned with the representation of properties of point clouds. Point clouds are described as multi-sets (unordered tuples) $X = [x_i]_i$ where each particle $x_i$ belongs to a configuration domain $\Omega$. We denote the set of all such multi-sets by $\mathrm{msets}(\Omega)$. For example, in molecular modeling, $x_i$ might describe the position and species of an atom and therefore $x_i = (\boldsymbol{r}_i, Z_i) \in \mathbb{R}^3 \times \mathbb{Z}$, while in high energy physics, one commonly uses the four-momentum $x_i = (E_i, \boldsymbol{p}_i) \in \mathbb{R}^4$, but one could also include additional features such as charge, spin, and so forth. A property of the point cloud is a map

$$\Phi\colon \mathrm{msets}(\Omega) \to Z \tag{1}$$

i.e., $X \mapsto \Phi(X) \in Z$, usually a scalar or tensor. The range space $Z$ is application dependent and left abstract throughout this paper. Expressing the input as a multi-set implicitly entails two important facts: (1) it can have varying lengths; (2) it is invariant under the permutations of the particles. The methods developed in this article are also applicable to fixed-length multi-sets, in which case $\Phi$ is simply a permutation-invariant function defined on some $\Omega^N$. Mappings that are not permutation-invariant are special cases with several simplifications.

In many applications, especially in the natural sciences, particle properties satisfy additional symmetries. When a group $G$ acts on $\Omega$ as well as on $Z$ we say that $\Phi$ is $G$-**equivariant** if

$$\Phi \circ g = \rho_Z(g)\Phi, \qquad g \in G \tag{2}$$

where $\rho_Z(g)$ is the action of the group element $g$ on the range space $Z$. In order to effectively incorporate exact group symmetry into properties $\Phi$, we consider model architectures of the form

$$\Phi\colon \mathrm{msets}(\Omega) \underset{\mathrm{embedding}}{\longrightarrow} V \underset{\mathrm{parameterization}}{\longrightarrow} V \underset{\mathrm{readout}}{\longrightarrow} Z, \tag{3}$$

where the space $V$ into which we "embed" the parameterization is a possibly infinite-dimensional vector space in which a convenient representation of the group is available. For simplicity we will sometimes assume that $Z = V$.

The Atomic Cluster Expansion (ACE) framework (Drautz, 2019; Dusson *et al.*, 2022; Drautz, 2020)) produces a complete linear basis for the space of all "smooth" $G$-equivariant properties $\Phi$ for the specific case when $G = \mathrm{O}(3)$ and $x_i$ are vectorial interatomic distances. Aspects of the ACE framework were incorporated into E(3)-equivariant message passing architectures, with significant improvements in accuracy (Batatia *et al.*, 2022a). In the following paragraphs we demonstrate that these ideas readily generalize to arbitrary reductive Lie groups.

## 4.1 Efficient many-body expansion

The first step is to expand $\Phi$ in terms of body orders, and truncate the expansion at a finite order $N$:

$$\Phi^{(N)}(X) = \varphi_0 + \sum_i \varphi_1(x_i) + \sum_{i_1, i_2} \varphi_2(x_{i_1}, x_{i_2}) + \cdots + \sum_{i_1, \ldots, i_N} \varphi_N(x_{i_1}, \ldots, x_{i_N}), \quad (4)$$

where $\varphi_n$ defines the $n$-body interaction. Formally, the expansion becomes systematic in the limit as $N \to \infty$. The second step is the expansion of the $n$-particle functions $\varphi_n$ in terms of a symmetrized tensor product basis. To define this we first need to specify the embedding of particles $x$: A countable family $(\phi_k)_k$ is a 1-particle basis if they are linearly independent on $\Omega$ and any smooth 1-particle function $\varphi_1$ (not necessarily equivariant) can be expanded in terms of $(\phi_k)_k$, i.e,

$$\varphi_1(x) = \sum_k w_k \phi_k(x). \quad (5)$$

For the sake of concreteness, we assume that $\phi_k : \Omega \to \mathbb{C}$, but the range can in principle be any field. We provide concrete examples of 1-particle bases in Appendix A.2. Let a complex vector space $V$ be given, into which the particle embedding maps, i.e.,

$$(\phi_k(x))_k \in V \qquad \forall x \in \Omega.$$

As a consequence of (5) any smooth scalar $n$-particle function $\varphi_n$ can be expanded in terms of the corresponding tensor product basis,

$$\varphi_n(x_1, \ldots, x_n) = \sum_{k_1, \ldots, k_n} w_{k_1 \ldots k_n} \prod_{s=1}^n \phi_{k_s}(x_s). \quad (6)$$

Inserting these expansions into (4) and interchanging summation (see appendix for the details) we arrive at a model for scalar permutation-symmetric properties,

$$A_k = \sum_{x \in X} \phi_k(x), \qquad \boldsymbol{A_k} = \prod_{s=1}^n A_k, \qquad \Phi^{(N)} = \sum_{\boldsymbol{k} \in \mathcal{K}} w_{\boldsymbol{k}} \boldsymbol{A_k}, \quad (7)$$

where $\mathcal{K}$ is the set of all $\boldsymbol{k}$ tuples indexing the features $\boldsymbol{A_k}$. Since $\boldsymbol{A_k}$ is invariant under permuting $\boldsymbol{k}$, only ordered $\boldsymbol{k}$ tuples are retained. The features $A_k$ are an embedding of $\mathrm{msets}(\Omega)$ into the space $V$. The tensorial product features (basis functions) $\boldsymbol{A_k}$ form a complete linear basis of multi-set functions on $\Omega$ and the weights $w_{\boldsymbol{k}}$ can be understood as a symmetric tensor. We will extend this linear cluster expansion model $\Phi^{(N)}$ to a message-passing type neural network model in § 4.4.

While the standard tensor product embeds $(\otimes_{s=1}^n \phi_{k_s})_{\boldsymbol{k}} : \Omega^n \to V^n$, the $n$-correlations $\boldsymbol{A_k}$ are *symmetric tensors* and embed $(\boldsymbol{A_k})_{\boldsymbol{k}} : \mathrm{msets}(\Omega) \to \mathrm{Sym}^n V$.

The evaluation of the symmetric tensor features $\boldsymbol{A_k}$ is the computational bottleneck in most scenarios, but efficient recursive evaluation algorithms (Batatia *et al.*, 2022a; Kaliuzhnyi and Ortner, 2022) are available. See Appendix A.13.2 for further discussion of model computational costs.

## 4.2 Symmetrisation

With (7) we obtained a systematic linear model for (smooth) multi-set functions. It remains to incorporate $G$-equivariance. We assume that $G$ is a reductive Lie group with a locally finite representation in $V$. In other words we choose a representation $\rho = (\rho_{kk'}) : G \to \mathrm{GL}(V)$ such that

$$\phi_k \circ g = \sum_{k'} \rho_{kk'}(g) \phi_{k'}, \quad (8)$$

where for each $k$ the sum over $k'$ is over a finite index-set depending only on $k$. Most Lie groups one encounters in physical applications belong to this class, the affine groups being notable exceptions. However, those can usually be treated in an *ad hoc* fashion, which is done in all $E(3)$-equivariant architectures we are aware of. In practice, these requirements restrict how we can choose the embedding $(\phi_k)_k$. If the point clouds $X = [x_i]_i$ are already given in terms of a representation of the group, then one may simply construct $V$ to be iterative tensor products of $\Omega$; see e.g. the MTP (Shapeev, 2016) and PELICAN (Bogatskiy *et al.*, 2022) models. To construct an equivariant two-particle basis we need to first construct the set of all intertwining operators from $V \otimes V \to V$. Concretely, we seek all solutions $C_{k_1 k_2}^{\boldsymbol{\alpha}, K}$ to the equation

$$\sum_{k_1' k_2'} C_{k_1' k_2'}^{\boldsymbol{\alpha}, K} \rho_{k_1' k_1}(g) \rho_{k_2' k_2}(g) = \sum_{K'} \rho_{KK'}(g) C_{k_1 k_2}^{\boldsymbol{\alpha}, K'}; \qquad (9)$$

or, written in operator notation, $C^{\boldsymbol{\alpha}} \rho \otimes \rho = \rho C^{\boldsymbol{\alpha}}$. We will call the $C_k^{\boldsymbol{\alpha}, K}$ *generalized Clebsch–Gordan coefficients* since in the case $G = \mathrm{SO}(3)$ acting on the spherical harmonics embedding $\phi_{lm} = Y_l^m$ those coefficients are exactly the classical Clebsch–Gordan coefficients. The index $\boldsymbol{\alpha}$ enumerates a basis of the space of all solutions to this equation. For the most common groups, one normally identifies a canonical basis $C^{\boldsymbol{\alpha}}$ and assigns a natural meaning to this index (cf. § A.5). Our abstract notation is chosen because of its generality and convenience for designing computational schemes. The generalization of the Clebsch–Gordan equation (9) to $n$ products of representations acting on the symmetric tensor space $\mathrm{Sym}^n(V)$ becomes (cf. § A.9)

$$\sum_{\boldsymbol{k}'} \mathcal{C}_{\boldsymbol{k}'}^{\boldsymbol{\alpha}, K} \overline{\rho}_{\boldsymbol{k}' \boldsymbol{k}} = \sum_{K'} \rho_{KK'} \mathcal{C}_{\boldsymbol{k}}^{\boldsymbol{\alpha}, K'} \qquad \forall K, \quad \boldsymbol{k} = (k_1, \ldots, k_N), \quad g \in G,$$

$$\text{where} \qquad \overline{\rho}_{\boldsymbol{k}' \boldsymbol{k}} = \sum_{\substack{\boldsymbol{k}'' = \pi \boldsymbol{k}' \\ \pi \in S_n}} \rho_{\boldsymbol{k}'' \boldsymbol{k}} \qquad \text{and} \qquad \rho_{\boldsymbol{k}' \boldsymbol{k}} = \prod_{t=1}^{n} \rho_{k_t' k_t}. \qquad (10)$$

Due to the symmetry of the $(\boldsymbol{A}_{\boldsymbol{k}})_{\boldsymbol{k}}$ tensors $\mathcal{C}_{\boldsymbol{k}}^{\boldsymbol{\alpha}, K}$ need only be computed for ordered $\boldsymbol{k}$ tuples and the sum $\sum_{\boldsymbol{k}'}$ also runs only over ordered $\boldsymbol{k}$ tuples. Again, the index $\boldsymbol{\alpha}$ enumerates a basis of the space of solutions. Equivalently, (10) can be written in compact notation as $\mathcal{C}^{\boldsymbol{\alpha}} \overline{\rho} = \rho \mathcal{C}^{\boldsymbol{\alpha}}$. These coupling operators for $N$ products can often (but not always) be constructed recursively from couplings of pairs (9). We can now define the symmetrized basis

$$\boldsymbol{B}_{\boldsymbol{\alpha}}^{K} = \sum_{\boldsymbol{k}'} \mathcal{C}_{\boldsymbol{k}'}^{\boldsymbol{\alpha}, K} \boldsymbol{A}_{\boldsymbol{k}'}. \qquad (11)$$

The equivariance of (11) is easily verified by applying a transformation $g \in G$ to the input (cf § A.6).

**Universality:** In the limit as the correlation order $N \to \infty$, the features $(\boldsymbol{B}_{\boldsymbol{\alpha}}^{K})_{K, \boldsymbol{\alpha}}$ form a complete basis of smooth equivariant multi-set functions, in a sense that we make precise in Appendix A.7. Any equivariant property $\Phi_V : \Omega \to V$ can be approximated by a linear model

$$\Phi_V^K = \sum_{\boldsymbol{\alpha}} c_{\boldsymbol{\alpha}}^K B_{\boldsymbol{\alpha}}^K, \qquad (12)$$

to within arbitrary accuracy by taking the number of terms in the linear combination to infinity.

### 4.3 Dimension Reduction

The tensor product of the cluster expansion in (7) is taken on all the indices of the one-particle basis. Unless the embedding $(\phi_k)_k$ is very low-dimensional it is often preferable to "sketch" this tensor product. For example, consider the canonical embedding of an atom $x_i = (\boldsymbol{r}_i, Z_i)$,

$$\phi_k(x_i) = \phi_{znlm}(x_i) = \delta_{z Z_i} R_{nl}(r_i) Y_l^m(\hat{\boldsymbol{r}}_i).$$

Only the $(lm)$ channels are involved in the representation of $\mathrm{O}(3)$ hence there is considerable freedom in "compressing" the $(z, n)$ channels.

Following Darby *et al.* (2022) we construct a sketched $G$-equivariant cluster expansion: We endow the one-particle basis with an additional index $c$, referred to as the sketched channel, replacing the index $k$ with the index pair $(c, k)$, and renaming the embedding $(\phi_{ck})_{c,k}$. In the case of three-dimensional particles one may, for example, choose $c = (z, n)$. In general, it is crucial that the representation

remains in terms of $\rho_{k,k'}$, that is, (8) becomes $\phi_{ck} \circ g = \sum_{k'} \rho_{kk'}(g)\phi_{ck'}$. Therefore, manipulating only the $c$ channel does not change any symmetry properties of the architecture. Generalizing Darby *et al.* (2022), the $G$-TRACE (tensor-reduced ACE) basis then becomes

$$\boldsymbol{B}_{c\boldsymbol{\alpha}}^K = \sum_{\boldsymbol{k'}} C_{\boldsymbol{k'}}^{\boldsymbol{\alpha},K} \tilde{\boldsymbol{A}}_{c\boldsymbol{k'}}, \qquad \text{where} \tag{13}$$

$$\tilde{\boldsymbol{A}}_{c\boldsymbol{k}} = \prod_{t=1}^{n} \left( \sum_{c'} w_{cc'} \sum_{x \in X} \phi_{c'k_t}(x) \right). \tag{14}$$

This construction is best understood as an equivariance-preserving canonical tensor decomposition (Darby *et al.*, 2022). There are numerous variations, but for the sake of simplicity, we restrict our presentation to this one case.

**Universality:** Following the proof of Darby *et al.* (2022) one can readily see that the $G$-TRACE architecture inherits the universality of the cluster expansion, in the limit of decoupled channels $\#c \to \infty$. A smooth equivariant property $\Phi$ may be approximated to within arbitrary accuracy by an expansion $\Phi^K(X) \approx \sum_{c,\boldsymbol{\alpha}} c_{\boldsymbol{\alpha}}^K \boldsymbol{B}_{c,\boldsymbol{\alpha}}^K(X)$. Since the embedding $\tilde{A}_{ck}$ is learnable, this is a *nonlinear model*. We refer to § A.7 for the details.

### 4.4 G-MACE, Multi-layer cluster expansion

The $G$-equivariant cluster expansion is readily generalized to a multi-layer architecture by re-expanding previous features in a new cluster expansion (Batatia *et al.*, 2022b). The multi-set $X$ is endowed with extra features, $\boldsymbol{h}_i^t = (h_{i,cK}^t)_{c,K}$, that are updated for $t \in \{1, ..., T\}$ iterations. These features themselves are chosen to be a field of representations such that they have a well-defined transformation under the action of the group. This results in

$$x_i^t = (x_i, \boldsymbol{h}_i^t) \tag{15}$$

$$\phi_{ck}^t(x_i, \boldsymbol{h}_i^t) = \sum_{\boldsymbol{\alpha}} w_{\boldsymbol{\alpha}}^{t,ck} \sum_{k',k''} C_{k'k''}^{\boldsymbol{\alpha},k} h_{i,ck'}^t \phi_{ck''}(x_i) \tag{16}$$

The recursive update of the features proceeds as in a standard message-passing framework but with the unique aspect that messages are formed via the $G$-TRACE and in particular can contain arbitrary high correlation order:.

$$m_{i,cK}^t = \sum_{\boldsymbol{\alpha}} W_{\boldsymbol{\alpha}}^{t,cK} \boldsymbol{B}_{c\boldsymbol{\alpha}}^{t,K}. \tag{17}$$

The gathered message $\boldsymbol{m}_i^t = (m_{i,cK}^t)_{c,k}$ is then used to update the particle states,

$$x_i^{t+1} = (x_i, \boldsymbol{h}_i^{t+1}), \qquad \boldsymbol{h}_i^{t+1} = U_t(\boldsymbol{m}_i^t), \tag{18}$$

where $U_t$ can be an arbitary fixed or learnable transformation (even the identity). Lastly, a readout function maps the state of a particle to a target quantity of interest, which could be *local* to each particle or *global* to the mset $X$,

$$y_i = \sum_{t=1}^{T} \mathcal{R}_t^{\text{loc}}(x_i^t), \qquad \text{respectively,} \qquad y = \sum_{t=1}^{T} \mathcal{R}_t^{\text{glob}}(\{x_i^t\}_i). \tag{19}$$

This multi-layer architecture corresponds to a general message-passing neural network with arbitrary body order of the message at each layer. We will refer to this architecture as $G$-MACE. The $G$-MACE architecture directly inherits universality from the $G$-ACE and $G$-TRACE architectures:

**Theorem 4.1** (Universality of $G$-MACE). *Assume that the one-particle embedding $(\phi_k)_k$ is a complete basis. Then, the set of G-MACE models, with a fixed finite number of layers $T$, is dense in the set of continuous and equivariant properties of point clouds $X \in \text{msets}(\Omega)$, in the topology of pointwise convergence. It is dense in the uniform topology on compact and size-bounded subsets.*

## 5 `lie-nn` : Generating Irreducible Representations for Reductive Lie Groups

In order to construct the G-cluster expansion for arbitrary Lie groups, one needs to compute the generalized Clebsch–Gordan coefficients (10) for a given tuple of representations (see 11). To facilitate this task, we have implemented an open source software library, `lie-nn`[1]. In this section we review the key techniques employed in this library.

---

[1] `https://github.com/xxx/xxx`

## 5.1 Lie Algebras of Reductive Lie Groups

Formally, the Lie algebra of a Lie group is its tangent space at the origin and carries an additional structure, the Lie bracket. Informally the Lie algebra can be thought of as a linear approximation to the Lie group but, due to the group structure, this linear approximation carries (almost) full information about the group. In particular the representation theory of the Group is almost entirely determined by the Lie algebra, which is a simpler object to work with instead of the fully nonlinear Lie group.

**Lie algebra** The Lie groups we study can be realized as closed subgroups $G \subset \mathrm{GL}_n(\mathbb{R})$ of the general linear group. In that case their Lie algebras can be concretely realized as $\mathfrak{g} = \mathrm{Lie}(G) = \{X \in M_n(\mathbb{R}) \mid \forall t \in \mathbb{R} : \exp(tX) \in G\}$ where $\exp(X) = 1 + X + \frac{1}{2}X^2...$ is the standard matrix exponential. It turns out that $\mathfrak{g} \subset M_n(\mathbb{R})$ is a linear subspace closed under the commutator bracket $[X, Y] = XY - YX$.

**Structure theory** We fix a linear basis $\{X_i\} \subset \mathfrak{g}$, called a set of generators for the group. The Lie algebra structure is determined by the *structure constants* $A_{ijk}$ defined by $[X_i, X_j] = \sum_k A_{ijk} X_k$, in that if $X = \sum_i a_i X_i$ and $Y = \sum_j b_j X_j$ then $[X, Y] = \sum_k \left( \sum_{i,j} A_{ijk} a_i b_j \right) X_k$. The classification of reductive groups provides convenient generating sets for their Lie algebras (or their complexifications). One identifies a large commutative subalgebra $\mathfrak{h} \subset \mathfrak{g}$ (sometimes of $\mathfrak{g}_{\mathbb{C}} = \mathfrak{g} \otimes_{\mathbb{R}} \mathbb{C}$) with basis $\{H_i\}$ so that most (or all) of the other generators $E_\alpha$ can be chosen so that $[H_i, E_\alpha] = \alpha(H_i) E_\alpha$ for a linear function $\alpha$ on $\mathfrak{h}$. These functions are the so-called *roots* of $\mathfrak{g}$. Structural information about $\mathfrak{g}$ is commonly encoded pictorially via the *Dynkin diagram* of $\mathfrak{g}$, a finite graph the nodes of which are a certain subset of the roots. There are four infinite families of simple complex Lie algebras $A_n = \mathfrak{su}(n+1), B_n = \mathfrak{so}(2n+1), C_n = \mathfrak{sp}(2n), D_n = \mathfrak{so}(2n)$ and further five exceptional simple complex Lie algebras (a general reductive Lie algebra is the direct sum of several simple ones and its centre). The Lie algebra only depends on the connected component of $G$. thus when the group $G$ is disconnected in addition to the infinitesimal generators $\{X_i\}$ one also needs to fix so-called "discrete generators", a subset $\mathbf{H} \subset G$ containing a representative from each connected component.

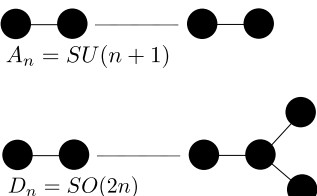

Figure 2: Examples of Dynkin diagrams and their associated group class.

**Representation theory** The representation theory of complex reductive Lie algebras is completely understood. Every finite-dimensional representation is (isomorphic to) the direct sum of irreducible representations ("irreps"), with the latter parametrized by appropriate linear functional on $\mathfrak{h}$ ("highest weight"). Further given a highest weight $\lambda$ there is a construction of the associated irrep with an explicit action of the infinitesimal generators chosen above. The **Weyl Dimension Formula** gives the dimension of an irrep in terms of its highest weight.

## 5.2 Numerical Computations in `lie-nn`

The most basic class of the `lie-nn` library encodes a group $G$ and infinitesimal representation $d\rho$ of $\mathfrak{g}$ using the tuple

$$\rho := (A, n, \{d\rho(X_i)\}_i, \{\rho(h)\}_{h \in \mathbf{H}}) , \tag{20}$$

with $A$ the structure constants of the group, $n$ the dimension of the representation, and $d\rho(X_i)$ and $\rho(h)$ being $n \times n$ matrices encoding the action of the infinitesimal and the discrete generators respectively. The action of infinitesimal generators is related to the action of group generators by the exponential, $\forall X \in \mathfrak{g}, \rho(e^X) = e^{d\rho(X)}$. For finite groups, we assume that $d\rho(X) = \mathbf{0}$ as they have only discrete generators.

As the building blocks of the theory irreps are treated specially; the package implements functionality for the following operations for each supported Lie group:

- Constructing the irrep with a given highest weight.

- Determining the dimension of an irrep.
- Decomposing the tensor product of several irreps into irreps up to isomorphism (the **selection rule**, giving the list of irreducible components and their multiplicities).
- Decomposing the tensor product of several irreps into irreps explicitly via a change of basis ("generalized **Clebsch–Gordan** coefficients").
- Computating the symmetrized tensor product of the group (see. 5.3 and A.9 for details).

To construct an irrep explicitly as in (20) one needs to choose a basis in the abstract representation space (including a labeling scheme for the basis) so that we can give matrix representations for the action of generators. For this purpose, we use in `lie-nn` the Gelfand-Tsetlin (GT) basis (Gelfand and Tsetlin, 1950) and associated labeling of the basis by GT patterns (this formalism was initially introduced for algebras of type $A_n$ but later generalized to all classical groups). Enumerating the GT patterns for a given algebra gives the dimension of a given irrep, the selection rules can be determined combinatorially, and it is also possible to give explicit algorithms to compute Clebsch–Gordan coefficients (the case of $A_n$ is treated by Alex *et al.* (2011)). For some specific groups, simplifications to this procedure are possible and GT patterns are not required.

In some cases, one wants to compute coefficients for reducible representations or for representations where the analytical computation with GT patterns is too complex. In these cases, a numerical algorithm to compute the coefficients is required. Let $d\rho_1, d\rho_2$ be two Lie aglebra representations of interest. The tensor product on the Lie algebra $d\rho_1 \otimes d\rho_2(X)$ can be computed as,

$$d\rho_1 \otimes d\rho_2 \ (X) = d\rho_1(X) \otimes 1 + 1 \otimes d\rho_2(X) \tag{21}$$

Therefore, given sets of generators of three representations $d\rho_1, d\rho_2, d\rho_3$, the Clebsch–Gordan coefficients are the change of basis between $(d\rho_1(X) \otimes 1 + 1 \otimes d\rho_2(X))$ and $d\rho_3(X)$. One can compute this change of basis numerically via a null space algorithm. For some groups, one can apply an iterative algorithm that generates all irreps starting with a single representation, using the above-mentioned procedure (see A.10).

### 5.3 Symmetric Powers

Let $V$ be a vector space and $\{e_i\}$ be a basis of $V$. The symmetric power of $V$, $\text{Sym}^n V$, can be regarded as the space of homogeneous polynomials of degree $n$ in the variables $e_i$. The product basis in Equation 10 spans exactly this space. A basis of $\text{Sym}^n V$ can be constructed as,

$$\{e_{i_1} \cdot e_{i_2} \cdot ... \cdot e_{i_n} | i_1 \leq ... \leq i_n\} \tag{22}$$

If $V_\lambda$ if an irreducible representation of a reductive Lie group $G$ with highest weight $\lambda$, $\text{Sym}^n V_\lambda$ admits a decomposition into irreducible representations,

$$\text{Sym}^n V_\lambda = \bigoplus c_{\lambda,\mu} V_\mu \tag{23}$$

The generalized Clebsch Gordan coefficients in (11) represent the change of basis between $\text{Sym}^n V_\lambda$ and one of the $V_\mu$. The following steps are taken to obtain these coefficients:

- Construct the symmetric power basis as in (22)
- Compute the coefficients $c_{\lambda,\mu}$, using Freudenthal's Formula or GT patterns.
- For any $\mu$ with $c_{\lambda,\mu}$ non-zero, find a basis of $V_\mu$, and compute the change of basis between the basis of $\text{Sym}^n V_\lambda$ and $V_\mu$

Alternatively, if one simply has the Clebsch Gordan coefficients, the change of basis from $V_\lambda$ to some $V_\mu$, a new algorithm outlined in Appendix A.9 and implemented in `lie-nn` can construct the change of basis from $\text{Sym}^n V_\lambda$ to $V_\mu$.

## 6 Applications

### 6.1 Lie groups and their applications

In Table 6.1 we give a non-exhaustive overview of Lie groups and their typical application domains, to which our methodology naturally applies.

Benchmarking our method on all of these applications is beyond the scope of the present work, in particular, because most of these fields do not have standardized benchmarks and baselines to

Table 1: Lie groups of interests covered by the present methods and their potential applications to equivariant neural networks. The groups above the horizontal line are already available in `lie-nn`. The ones below the line fall within our framework and can be added.

| Group | Application | Reference |
|---|---|---|
| U(1) | Electromagnetism | (Lagrave *et al.*, 2021) |
| SU(3) | Quantum Chromodynamics | (Favoni *et al.*, 2022) |
| SO(3) | 3D point clouds | (Batatia *et al.*, 2022a) |
| $SO^+(1,3)$ | Particle Physics | (Bogatskiy *et al.*, 2020b) |
| $SL(3, \mathbb{R})$ | Point cloud classification | - |
| $SU(2^N)$ | Entangled QP | - |
| Sp(N) | Hamiltonian dynamics | - |
| SO(2N + 1) | Projective geometry | - |

compare against. The MACE architecture has proven to be state of the art for a large range of atomistic modeling benchmarks (Batatia *et al.*, 2022a). In the next section, we choose two new prototypical applications and their respective groups to further assess the performance of our general approach.

## 6.2 Particle physics with the $SO(1,3)$

Jet tagging consists in identifying the process that generated a collimated spray of particles called a *jet* after a high-energy collision occurs at particle colliders. Each jet can be defined as a multiset of four-momenta $[(E_i, \mathbf{p}_i)]_{i=1}^N$, where $E_i \in \mathbb{R}^+$ and $\mathbf{p}_i \in \mathbb{R}^3$.

Current state-of-the-art models incorporate the natural symmetry arising from relativistic objects, e.g, the Lorentz symmetry, as model invariance. To showcase the performance and generality of the $G$-MACE framework we use the Top-Tagging dataset (Butter *et al.*, 2019), where the task is to differentiate boosted top quarks from the background composed of gluons and light quark jets. In Table 6.2, we can see that $G$-MACE achieves excellent accuracy, being the only arbitrary equivariant model to reach similar accuracy as PELICAN, which is an invariant model. We refer to Appendix A.11.1 for the details of the architecture.

Table 2: Comparisson between state-of-the-art metrics on the Top-Tagging dataset. Scores were taken from (Bogatskiy *et al.*, 2022; Qu *et al.*, 2022; Qu and Gouskos, 2020; Munoz *et al.*, 2022; Bogatskiy *et al.*, 2020a; Komiske *et al.*, 2019; Pearkes *et al.*, 2017).

| Architecture | #Params | Accuracy | AUC | Rej$_{30\%}$ |
|---|---|---|---|---|
| **PELICAN** | 45k | **0.942** | **0.987** | **2289 ± 204** |
| **partT** | 2.14M | 0.940 | 0.986 | 1602 ± 81 |
| **ParticleNet** | 498k | 0.938 | 0.985 | 1298 ± 46 |
| **LorentzNet** | 224k | **0.942** | **0.987** | 2195 ± 173 |
| **BIP** | 4k | 0.931 | 0.981 | 853 ± 68 |
| **LGN** | 4.5k | 0.929 | 0.964 | 435 ± 95 |
| **EFN** | 82k | 0.927 | 0.979 | 888 ± 17 |
| **TopoDNN** | 59k | 0.916 | 0.972 | 295 ± 5 |
| **LorentzMACE** | 228k | **0.942** | **0.987** | 1935 ± 85 |

## 6.3 3D Shape recognition

3D shape recognition from point clouds is of central importance for computer vision. We use the ModelNet10 dataset (Wu *et al.*, 2015) to test our proposed architecture in this setting. As rotated objects need to map to the same class, we use a MACE model with $O(3)$ symmetry. To create an encoder version of $G$-MACE, we augment a PointNet++ implementation (Yan, 2019) with $G$-MACE layers. See the appendix A.11.2 for more details on the architecture. We see in Table 3 that MACE outperforms the non-equivariant baseline.

Table 3: Accuracy in shape recognition. Scores were taken from (Qi *et al.*, 2016), (Qi *et al.*, 2017), Shen *et al.* (2018), Li *et al.* (2018), Kumawat and Raman (2019)

| Architecture | **PointMACE** (ours) | **PointNet** | **PointNet ++** | **KCN** | **SO-Net** | **LP-3DCNN** |
|---|---|---|---|---|---|---|
| Accuracy | **96.1** | 94.2 | 95.0 | 94.4 | 95.5 | 94.4 |
| Representation | Point Cloud | Point cloud | Point cloud | Point Cloud | Point Cloud | Voxel grid |

## 7 Conclusion

We introduced the *G*-Equivariant Cluster Expansion, which generalizes the successful ACE and MACE architectures to symmetries under arbitrary reductive Lie groups. We provide an open-source Python library `lie-nn` that provides all the essential tools to construct such general Lie-group equivariant neural networks. We demonstrated that the general *G*-MACE architecture simultaneously achieves excellent accuracy in Chemistry, Particle Physics, and Computer Vision. Future development will implement additional groups and generalize to new application domains.

## Acknowledgments and Disclosure of Funding

IB's work was supported by the ENS Paris Saclay. CO's work was supported by NSERC Discovery Grant IDGR019381 and NFRF Exploration Grant GR022937. This work was also performed using resources provided by the Cambridge Service for Data Driven Discovery (CSD3).IB would like to thank Gábor Csányi for his support.

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

# A   Appendix

## A.1   Background on Lie groups and Tensor Products of Representations

### A.1.1   Lie groups

Lie groups generalize the idea of continuous symmetry such as the rotation symmetry of the round sphere $S^n$ (given by the action of the orthogonal group $\mathrm{O}(n)$), the symmetry of a vector space under changes of basis (given by the action of the general linear group $\mathrm{GL}_n(\mathbb{R})$), or the invariance of Minkowski spacetime under boosts (given by the Lorentz group). The symmetry groups of specific geometries were studied in the 19th century, leading to Sophus Lie's study of general continuous symmetry. Other major contributors to our understanding include Klein, Riemann, Hilbert, Poincarré, Noether, Cartan, among many others. In particular the solution of Hilbert's 5th problem by Gleason, Montgomery, and Zippin shows that under very general conditions a continuous symmetry group must be a Lie group. Symmetry being central to mathematics Lie groups appearl in many areas of mathematics, from differential geometry to number theory.

Let us give the general formal definition, as well as a more practical concrete one:

**Definition A.1.** A real (complex) **Lie group** is a group that is also a finite-dimensional smooth (complex) manifold in which the product and inversion of the group are also smooth (holomorphic) maps.

A **linear Lie group** is a subgroup of the group $\mathrm{GL}_n(\mathbb{R})$ which is also a closed subset (in the sense of real analysis).

It is a fact that every closed subgroup of a Lie group is a Lie group, so every group of the second type is indeed also a group of the first type. Conversely while in general a Lie group is only *locally* isomorphic to a linear group (by a surjective group homomorphism), every compact Lie group is isomorphic to a linear group, and most Lie groups that arise in practice are linear.

When the Lie group $G$ acts on a space $X$ (e.g. as a symmetry group) it naturally also acts on spaces of functions on $X$ (by translation), and this action is *linear*. In general. an action of $G$ on a (suually complex) vector space by linear map is called a **linear representation** (of $G$). In the context of our paper the space of functions on $X$ is the space of features, and we achieve efficient computation in it by decomposing this representation into simpler constituents – in other words by understanding the **representation theory** of $G$.

Describing the representation theory of arbitrary Lie groups is a very difficult problem and an active area of research in pure mathematics. However, for certain classes of groups it is possible to say quite a bit (for example classical harmonic analysis can be viewed from the lens of the representation theory of *commutative* groups). An important class of groups whose representation theory can be completely understood is the class of *reductive* Lie groups, on which we focus in this paper.

For now let us give a concrete definition; an abstract will follow later once we introduce the necessary language.

**Definition A.2.** A linear Lie group $G \subset \mathrm{GL}_n(\mathbb{R})$ is **reductive** if it is also closed under transpose: for every $g \in G$ we have that $g^T \in G$.

Most of the groups that are of interest in natural sciences are reductive Lie groups. This includes well-known groups such as the symplectic group $\mathrm{Sp}_{2n}(\mathbb{R})$ relevant in Hamiltonian dynamics, and orthogonal group $\mathrm{O}(n)$ which parametrizes rotations in $\mathbb{R}^n$.

### A.1.2   Linear representations

Fix a Lie group $G$. A **linear representation** of $G$ is a pair $(\pi, V)$ where $V$ is a complex vecctor space and $\pi$ is an action of $G$ on $V$ by linear maps. In other words for each $g \in G$ we have a complex-linear map $\pi(g)\colon V \to V$ such that $\pi(g_1 g_2) = \pi(g_1)\pi(g_2)$ and such that $\pi(1_G) = \mathrm{Id}_V$ (in other words, $\pi\colon G \to \mathrm{GL}(V)$ is a group homomorphism).

For an example let the familiar rotation group $\mathrm{O}(3)$ act on the usual round sphere $S^2$. Then $\mathrm{O}(3)$ acts on the space of functions on $S^2$. This space is infinite-dimensional, but it turns out we can approximate it by a sum of finite-dimensional pieces. Indeed for each degree $d \geq 0$ let $V_d$ be the space of polynomials in the variables $x, y, z$ which are homogeneous of degree $d$ (for example

$V_0 = \mathrm{span}\{1\}$, $V_1 = \mathrm{span}\{x, y, z\}$, $V_2 = \mathrm{span}\{x^2, xy, y^2, \ldots\}$. Letting $\mathrm{O}(3)$ act on $\mathbb{R}^3$ by linear change of variable, it acts on each $V_d$ since such change of variable does not change the degree of homogeneity. Now let $W_d \subset V_d$ be the subspace of **harmonic** polynomials, those polynomials $p$ that satisfy $\frac{\partial^2 p}{\partial x^2} + \frac{\partial^2 p}{\partial y^2} + \frac{\partial^2 p}{\partial z^2} = 0$. Then $W_d$ is itself $\mathrm{O}(3)$ invariant since the Laplace operator is rotation-invariant, making $W_d$ a **subrepresentation** of $V_d$. It is a (not immediately obvious) fact that each $W_d$ is in fact an **irreducible** representation: it has no subrepresentations of its own other than itself and the zero subspace. Further, if we interpret the harmonic polynomials in $W_d$ as functions on the unit sphere $S^2 \subset \mathbb{R}^3$ we can view $W_d$ as a space of functions on the sphere. It then turns out that the spaces $W_d$ are linearly independent of eaceh other, and that their sum $\bigoplus_{d \geq 0} W_d$ is dense in any resaonable space of functions on the sphere (the space of continuous functions, the space of smooth functions, or the space of square-integrable functions, each with its respective notion of convergence). In other words, we have morally decomposed the space of functions on $S^2$ as the sum of $\mathrm{O}(3)$-invariant subrepresentations, each of which is irreducible.

Decomposing a representation into its irreducible constituents is a major goal of representation theory (as well as a computational goal of this paper). When this is possible we can understand a general representations $V$ of a group $G$ by first enumerating its irreducible representations and then counting how many times each irreducible occurs as summand in $V$, known as the (this is very much analogous to understanding general positive integers via their prime factorization – here the irreducible representations play the role of the prime numbers).

Reductive groups are particularly amenable to this approach because their representations indeed do decompose as direct sums in this fashion. It is fact that a linear Lie group $G$ is reductive if and only if each of its finite-dimensional representations is isomorphic to a direct sum of irreducible representations.

Before we delve further into representation theory we need a brief detour into the structure theory of Lie groups.

### A.1.3 Lie algebras

As Lie groups are manifolds, understanding their representations theory usually leads to advanced geometrical and topological results. We will now describe how these difficult questions can be transformed into simpler linear algebra questions using the Lie algebras associated with Lie groups.

A **Lie algebra** is a vector space $\mathfrak{g}$ over a field $F$ together with a bilinear map $[\cdot, \cdot] : \mathfrak{g} \times \mathfrak{g} \to \mathfrak{g}$ called the Lie bracket, satisfying the following axioms:

- The Lie bracket is bilinear, that is for all $x, y, z \in \mathfrak{g}$ and $\alpha, \beta \in F$, $[\alpha x + \beta y, z] = \alpha[x, z] + \beta[y, z]$ and $[x, \alpha y + \beta z] = \alpha[x, y] + \beta[x, z]$.
- The Lie bracket is antisymmetric, that is for all $x, y \in \mathfrak{g}$, $[x, y] = -[y, x]$.
- The Lie bracket satisfies the Jacobi identity, that is for all $x, y, z \in \mathfrak{g}$, $[x, [y, z]] + [y, [z, x]] + [z, [x, y]] = 0$.

Lie algebras can alternatively be seen as the tangent space of a Lie group to the identity:

**Theorem A.3.** *Let $G$ be a Lie group viewed as a smooth manifold, the tangent space $T_I G$ is a Lie algebra called the Lie algebra of $G$ noted $\mathfrak{g}$. Moreover in the case of connected complex Lie groups, the category of representation of $G$, $Rep(G)$, is equivalent to the category of representation of $\mathfrak{g}$, $Rep(\mathfrak{g})$.*

This theorem is at the heart of the representation theory of Lie groups. As Lie algebra are vector spaces, the functorial equivalence between the two categories of representation translates geometrical and topological questions into simpler linear algebra questions.

A classification of Lie algebras was carried out by Cartan and later Gelfand work. In the case of complex Lie algebras, this classification is complete. Two major classes of Lie algebras emerged from Cartan's work, semi-simple Lie algebras and nilpotent Lie algebras.

**Definition A.4** (Central series). A central series of a Lie algebra $\mathfrak{g}$ is a sequence of ideals $(\mathfrak{g}_i)_{i \in \mathbb{N}}$ such that $\mathfrak{g}_0 = \mathfrak{g}$ and $\mathfrak{g}_{i+1} = [\mathfrak{g}_i, \mathfrak{g}]$.

**Definition A.5.** A Lie algebra is called semi-simple if it does not contain any non-trivial ideals. A Lie algebra is called nilpotent if its central series terminates to zero.

An important result of this classification is that any Lie algebras over an algebrically closed field can be decomposed into semisimple parts and a nilpotent part. We call this the Jordan decomposition. Therefore, understanding arbitrary representations of a Lie algebra boils down to understanding representation of both semi-simple and nilpotent algebras.

Most reductive Lie groups of interest have a semi-simple Lie algebra as associated Lie algebra. These algebras are also the one for which the representation theory is the best understood.

The study of Lie algebras is decomposed in two subfields. The structure theory studies the classification of Lie algebras in terms of their eigenspaces (as vector spaces). The representation theory builds on the structure theory to classify representations of semi-simple Lie algebras. We will give a very short introduction to both of them.

### A.1.4 Structure theory of semi-simple Lie algebras

Let $\mathfrak{g}$ be a semi-simple complex Lie algebras. Let $\mathfrak{h}$ be one maximal set of mutually commutative element of $\mathfrak{g}$. Maximal means here that no other such set contains strictly more element than $\mathfrak{h}$. We call $\mathfrak{h}$ the Cartan algebra of $\mathfrak{g}$. As elements of $\mathfrak{h}$ are mutually commutative, they share the same eigenspaces.

Therefore one can decompose $\mathfrak{g}$ in terms of eigenspaces of elements of $\mathfrak{h}$. Let $\alpha \in \mathfrak{h}^*$ be a function in the dual of the Cartan algebra such that $\forall h_i \in \mathfrak{h}, \alpha(h_i) = \alpha_i$ with $\alpha_i$ the eigenvalue associated to $h_i$. We call this linear operator the roots of $\mathfrak{g}$. Let $R$ be the space of all this functions, called the root space. One therefore has the the following decomposition of the Lie algebra;

$$\mathfrak{g} = \mathfrak{h} + \bigoplus_{\alpha \in R} g_\alpha \tag{24}$$

The structure theory of semi-simple complex Lie algebras is the study of the root space $R$ and the associated eigenspaces $g_\alpha$. One can clasify complex semi-simple Lie algebras based on their type of root spaces. For real semi-simple Lie algebras, the structure theory essential reduces down to the complex case with a few extra steps.

### A.1.5 Representation theory of complex semi-simple Lie algebras and Lie groups

The representation theory of complex reductive Lie algebras is completely understood. Every finite-dimensional representation is the direct sum of irreducible representations. Every irreducible representation is uniquely determined by a highest weight $\lambda$, which is an element of the dual of the Cartan algebra $\mathfrak{h}$. The dimension of each irreducible representation can be computed using the Weyl dimension formula.

$$\dim V_\lambda = \frac{\prod_{\alpha \in R^+} (\lambda + \rho, \alpha)}{\prod_{\alpha \in R^+} (\rho, \alpha)} \tag{25}$$

where $R^+$ is a subset of the root space $R$, called the positive roots, and $\rho$ is the half sum of the positive roots. Obtaining explicitly a basis for each representation is much harder. Gelfand and Tselin proposed a basis for the general case of $SU(N)$ groups. This basis was recently generalized to other complex semi-simple algebras Molev (1999).

### A.1.6 Representations of real semi-simple Lie groups

The representation theory of real semi-simple Lie groups can be elucidated by examining the complex semi-simple case through the application of the so-called Weyl unitary trick. This technique establishes a connection between the representations of real Lie groups and those of the complexification of their universal covering groups.

**Definition A.6.** Given a Lie algebra $\mathfrak{g}$, there exists a unique simply connected group $\hat{G}$ possessing $\mathfrak{g}$ as its Lie algebra. A group homomorphism $\psi$ exists from this unique simply connected Lie group to any group $G$ sharing the same Lie algebra. We refer to $\hat{G}$ as the universal covering Lie group, and it is unique up to isomorphism.

An essential observation is that the irreducible representations of $G$ form a subset of the irreducible representations of $\hat{G}$. Understanding the irreducible representations of $\hat{G}$ suffices. Take the case of

the $\mathfrak{su}(2)$ Lie algebra, for instance. The universal covering group is $\mathrm{SU}(2)$, corresponding to the special unitary two-by-two matrices. The group $\mathrm{SO}(3)$ of special orthogonal matrices also has $\mathfrak{su}(2)$ as its Lie algebra. Consequently, the irreducible representations of $\mathrm{SO}(3)$ are a subset of those of $\mathfrak{su}(2)$. Representations of $\mathfrak{su}(2)$ can be indexed by a half-integer, known as the quantum number $j$. The integer representations are also irreducible representations of $\mathrm{SO}(3)$, with non-integer ones being referred to as the spin representations.

Since the representation theory of complex Lie groups is well understood, an optimal approach to understanding the representation theory of a real Lie group involves studying the complexification of its universal cover.

**Definition A.7.** For a universal covering Lie group $G$ with Lie algebra $\mathfrak{g}$, the universal covering Lie group $\hat{G}_{\mathbb{C}}$, having the associated Lie algebra $\mathfrak{g}_{\mathbb{C}} = \mathfrak{g} \otimes \mathbb{C}$, is termed the complexification of $\hat{G}$.

The group of interest $G$, its universal cover $\hat{G}$ and the complexification of the universal cover $\hat{G}_{\mathbb{C}}$, all share the similar irreducible representations. Therefore one obtains the following chain of inclusion of the respective representations of the groups,

$$Rep(G) \subset Rep(\hat{G}) \subset Rep(\hat{G}_{\mathbb{C}}) \tag{26}$$

Let $K$ be the maximal compact subgroup of $\hat{G}_{\mathbb{C}}$ and $\rho_\lambda$ a finite dimensional irreducible representation of $K$, with $\lambda$ as its highest weight. It is possible to analytically continue $\rho_\lambda$ into an irreducible representation of $\hat{G}_{\mathbb{C}}$, denoted as $\rho_\lambda^{\hat{G}_{\mathbb{C}}}$, and also into the analytical conjugate $\bar{\rho}_\lambda^{\hat{G}_{\mathbb{C}}}$. Every finite-dimensional representation of $\hat{G}_{\mathbb{C}}$ can be constructed as the tensor product form $\rho_\lambda^{\hat{G}_{\mathbb{C}}} \otimes \bar{\rho}_{\lambda'}^{\hat{G}_{\mathbb{C}}}$. Due to the fact that finite-dimensional representations of the universal covers and complexification of a semi-simple Lie group are isomorphic, one can express:

$$\rho_\lambda^{\hat{G}_{\mathbb{C}}} \otimes \bar{\rho}_{\lambda'}^{\hat{G}_{\mathbb{C}}} := \rho_{(\lambda,\lambda')}^{\hat{G}} := \rho_{(\lambda,\lambda')}^{G} \tag{27}$$

As universal covering groups contain more representations than the original cover, this mapping is not an isomorphism for all lambda. To illustrate this, consider the example of the Lorentz group $SO(1,3)$.
The universal cover of $\mathrm{SO}(1,3)$ is $\mathrm{SL}(2,C)$, and the maximal compact subgroup of $\mathrm{SL}(2,C)$ is $\mathrm{SU}(2)$. The irreducible representations of $\mathrm{SU}(2)$ are indexed by $l \in \mathbb{N}/2$ and correspond to the Wigner D matrices $D^l(g), g \in \mathrm{SU}(2)$. The $\mathrm{SU}(2)$ group elements are parameterized by Euler angles that can be analytically continued to $\mathrm{SL}(2,C)$ as follows:

$$\alpha = \phi + i\kappa, \beta = \theta + i\epsilon, \gamma = \phi + i\xi \\ \phi \in [0, 2\pi), \theta \in [0, \pi], \phi \in [0, 2\pi), \kappa, \epsilon, \xi \in \mathbb{R} \tag{28}$$

This enables the construction of the fundamental representation of $\mathrm{SL}(2,\mathbb{C})$, resulting in representations of the Lorentz group corresponding to the product of Wigner D matrices:

$$D^{k/2}(\alpha, \beta, \gamma) \otimes \bar{D}^{l/2}(\alpha, \beta, \gamma) \tag{29}$$

The irreducible representations of $\mathrm{SO}(1,3)$ are indexed by a pair of integers $(l, k)$ corresponding to the associated $SU(2)$ representations in the tensor product.

### A.1.7 Gefland-Tselin Basis

The enumeration of finite-dimensional representations of complex Lie groups has been well understood. However, the derivation of explicit matrix representations in specific bases and the tensor product decomposition of representations within these bases remain challenging and subject to ongoing research. The Gelfand-Tselin basis provides an explicit structure for irreducible representations, initially designed for the $SU(N)$ group Gelfand and Tsetlin (1950); Alex *et al.* (2011), and subsequently generalized to encompass other classical Lie groups Molev (1999).

Central to the concept of GT patterns is the application of branching rules, whereby representations of a larger group are constructed by enumerating induced representations on its subgroups. Consider, for instance, the $SU(N)$ case where irreducible representations are characterized by the highest weight vector $\lambda$, a vector of length $N$ terminated by a zero, $\lambda = (\lambda_1, \ldots, \lambda_{N-1}, 0)$. This highest weight

vector may spawn several induced representations of the subgroup $SU(N-1)$, each described by $N-1$ vectors. This process can be iteratively executed in a descending chain:

$$SU(N) \to SU(N-1) \to \cdots \to SU(2) \to U(1) \tag{30}$$

This chain commences with the group of interest and culminates with the smallest subgroup. From this branching chain, triangular arrays known as GT-patterns can be constructed, with each row corresponding to a vector of a representation ranging from $SU(N)$ to $U(1)$:

$$\begin{pmatrix} \lambda_{N,1} & \lambda_{N,2} & \cdots & \lambda_{N,N-1} & 0 \\ \lambda_{N-1,1} & \lambda_{N-1,2} & \cdots & \lambda_{N-1,N-1} & \\ \vdots & \vdots & & \cdot^{\cdot^{\cdot}} & \\ \lambda_{21} & \lambda_{22} & & & \\ \lambda_{11} & & & & \end{pmatrix} \tag{31}$$

Adjacent rows within these patterns must adhere to a so-called "snake rule" to form an admissible GT pattern:

$$\lambda_{k,1} > \lambda_{k-1,1} > \lambda_{k,2} > \lambda_{k-1,2} > \ldots > \lambda_{k,k-1} > \lambda_{k-1,k-1} > \lambda_{k,k} \tag{32}$$

The dimension of a specific irreducible representation corresponds to the count of such admissible arrays, with the top row representing the highest weight vector. Representations sharing the same highest weight vectors interrelate through ladder operators. One key attribute of GT patterns is that the action of ladder operators on a GT pattern is analytically known, enabling the construction of the matrix representation for all representations using only the highest weight representation. Furthermore, constructing the highest weight matrix representation equals solving a linear system when the ladder operators are known analytically. Extending this methodology to other classical Lie groups (such as $(n), \mathrm{Sp}(n), \mathrm{SL}(n)$) presents challenges as the ladder operators' entries become increasingly complex and recursive.

## A.2 Example of one particle basis for O(3) and SO(1,3) groups

In the paper, the one-particle basis $\phi$ is left abstract, as it depends on the specific type of input point cloud and the group under consideration. We now describe two concrete examples, one involving isometry the other Lorentz equivariance.

### A.2.1 Isometry equivariance

In the first example we consider a point cloud comprised of atoms, for example representing a molecule. Here, the point states are denoted by $x_i = (\boldsymbol{r}_i, Z_i) \in \mathbb{R}^3 \times \mathbb{Z}$, where $\boldsymbol{r}_i$ corresponds to the position in 3D and $Z_i$, its atomic number, is a categorical variable. The one-particle basis must form a complete basis for $\mathbb{R}^3 \times \mathbb{Z}$, which is both rotationally equivariant and translationally invariant. The translation is typically handled by referring to the atom's state in relation to a central value or another point. For simplicity, we can assume $\boldsymbol{r}_i$ the particle's position relative to the point cloud's centre of mass, ensuring translational invariance.

The space $\mathbb{R}^3$ can be represented as the product $\mathbb{R} \times S^2$, with $S^2$ being the sphere. As such, we choose a basis for $\mathbb{R}$ such as Bessel functions, Chebyshev polynomials or a Fourier basis, denoted as $R_n$. For $S^2$, we utilise the spherical harmonics $Y_{lm}$, which form a basis of functions in the sphere. Consequently, the one-particle basis can be expanded as:

$$\phi_{nlm}(x_j) = R_n(r_i) Y_{lm}(\hat{r}_i) f(Z_i) \tag{33}$$

where $r_i$ corresponds to the norm of $\boldsymbol{r}_i$ and $\hat{r}_i = \frac{\boldsymbol{r}_i}{r_i}$.

The case of the Lorentz group is more complex than the $(3)$ case. Consider a point cloud made up of elementary particles, represented by their 4-momenta $x_i = (E_i, \boldsymbol{p}_i) \in \mathbb{R}^4$. Although the 4-momenta is a vector, the natural metric in this space is not the Euclidean metric but rather a pseudo metric known as the Minkowski metric. This pseudo-metric encapsulates the unique role that time plays in special relativity. In natural units, the pseudo norm $\eta$ of a 4-vector $u = (u_0, u_1, u_2, u_3)$ is defined as

$$\eta(u, u) = u_0^2 - u_1^2 - u_2^2 - u_3^2 \tag{34}$$

To construct the one-particle basis, it is necessary to expand a four-vector in terms of the basis of representations of the Lorentz groups. As outlined in Appendix A.1.6, irreducible representations

of the Lorentz group are characterised by a tuple of integers $(l, k)$. The 4-vectors correspond to the fundamental, irreducible representation, namely the $(1, 1)$ representation. Analogous structures to the spherical harmonics can be obtained by examining the symmetric powers of the irreducible representations. These are analogous to harmonic polynomials on Minkowski space, much as spherical harmonics are harmonic polynomials on the sphere $S^2$. We thus define the Minkowski spherical harmonics as the following symmetric tensor product,

$$Y_{lm}(\mathbf{p}_i) = \text{Sym}^l \mathbf{p}_i \tag{35}$$

These spherical harmonics form a harmonic polynomial basis on Minkowski space. Therefore, one can create the one-particle basis in a manner akin to the $O(3)$ case, as follows:

$$\phi_{nlm}(\mathbf{p}_j) = R_n(p_i)Y_{lm}(\hat{p}_i) \tag{36}$$

Where $p_i = \eta(\boldsymbol{p}_i, \boldsymbol{p}_i)$, and $\hat{p}_i = \frac{\boldsymbol{p}_i}{p_i+\epsilon}$, with $\epsilon$ being a small constant that prevents divergence when $p_i$ is zero due to the non-positiveness of the Minkowski norm.

### A.3 Extended Related Work

Convolutional Neural Networks (CNNs), which are translation equivariance, initiated the utilization of data symmetry in machine learning architectures. Throughout time, CNNs have been extended to include other symmetries as well. Central to all these generalizations is the group averaging operation,

$$\text{Avg}(f)(x) = \int_{g \in G} f(g \cdot x)dg, \tag{37}$$

Where $x$ denotes the input signal or feature, $f$ is the convolution kernel, $G$ represents the group of interest, and $dg$ is an invariant measure on $G$. This transformation is essential, as it converts any convolution into a group invariant convolution. The feasibility of this approach largely depends on the computational simplicity of the integral. The most straightforward instance occurs for finite groups, as explained by G-convolutions (Cohen and Welling, 2016b), where the integral simplifies to a sum,

$$\text{Avg}(f)(x) = \sum_{g \in G} f(g \cdot x). \tag{38}$$

In the context of $G$ being a compact group, the invariant measure $dg$ is unique, referred to as the Haar measure of the group, allowing the integral to be computed numerically, e.g., on a grid. Steerable CNNs (Cohen and Welling, 2016a), LieConv Finzi *et al.* (2020) and Spherical CNNs (Cohen *et al.*, 2018) extended CNNs to $O(2)$ and $O(3)$ equivariance, respectively. A general theory for convolution on any compact group and symmetric space was developed by Kondor and Trivedi (2018), and further extended to equivariant convolutions on Riemannian manifolds by Weiler *et al.* (2021). However, this approach has several limitations,

- The direct computation of the integral can be numerically unstable and inefficient, even for relatively small groups like $O(3)$.
- In the case of non-compact groups, a unique invariant measure is absent, and the integral diverges.
- Across these methods, the convolution kernel $f$ is usually constrained to a two-body operator .

In the case of compact groups, the integral over the group may be calculated via an alternative means. There exists a linear operator, called the Clebsch Gordan operator $\mathcal{C}$, such that,

$$\text{Avg}(f)(x) = \mathcal{C}(f)(x) \tag{39}$$

Therefore, the complex integral over the group becomes a linear operation. The form of this operator depends on the basis in which $f$ is expended. In the case of $G = O(3)$ and if this basis is carefully chosen, the entries of this operator are known analytically. This approach is numerically stable and more efficient and was taken by numerous works, including ACE Dusson *et al.* (2022), Cormorant Anderson *et al.* (2019), NequIP Batzner *et al.* (2022), or MACE Batatia *et al.* (2022b). The central aim of our work is to show that this approach can also be generalized to all reductive Lie groups, even non-compact ones, and provide tools to do so.

## A.4 Proof of (7)

This statement follows closely the arguments by Dusson *et al.* (2022); Drautz (2020) and others.

$$\sum_{j_1,\dots,j_n} \sum_{\boldsymbol{k}} w_{\boldsymbol{k}} \prod_s \phi_{k_s}(x_{j_s}) = \sum_{\boldsymbol{k}} w_{\boldsymbol{k}} \sum_{j_1,\dots,j_n} \prod_s \phi_{k_s}(x_{j_s})$$

$$= \sum_{\boldsymbol{k}} w_{\boldsymbol{k}} \prod_{s=1}^{n} \sum_j \phi_{k_s}(x_j)$$

$$= \sum_{\boldsymbol{k}} w_{\boldsymbol{k}} \prod_{s=1}^{n} A_k$$

$$= \sum_{\boldsymbol{k}} w_{\boldsymbol{k}} \boldsymbol{A}_{\boldsymbol{k}}.$$

## A.5 Custom notation and indexing

We briefly contrast our notation for Clebsch–Gordan coefficients (10) with the standard notation. By means of example, consider the group $SO(3)$ in which case the Clebsch–Gordan equations are written as

$$\sum_{m_1' m_2'} C^{LM}_{l_1 m_1' l_2 m_2'} \rho^{l_1}_{m_1' m_1}(g) \rho^{l_2}_{m_2' m_2}(g) = \sum_{M'} \rho^{L}_{MM'}(g) C^{LM'}_{l_1 m_1 l_2 m_2}. \tag{40}$$

In this setting, our index $\boldsymbol{\alpha}$ simply enumerates all possible such coefficients. One can often assign a natural meaning to this index, e.g., for the group $SO(3)$ it is given by the pair of angular quantum numbers $(l_1, l_2)$. Specifically, in this case, we obtain

$$C^{\boldsymbol{\alpha},LM}_{l_1 m_1 l_2 m_2} = \begin{cases} C^{LM}_{l_1 m_1 l_2 m_2}, & \text{if } \boldsymbol{\alpha} = (l_1, l_2), \\ 0, & \text{otherwise}, \end{cases} \tag{41}$$

where $C^{LM}_{l_1 m_1 l_2 m_2}$ are the Clebsch–Gordan coefficients in the classical notation. Thus, the additional index $\boldsymbol{\alpha}$ is not really required in the case of $SO(3)$, nor our other main example, $SO(1,3)$. Our notation is still useful to organize the computations of equivariant models, especially when additional channels are present, which is usually the case. Moreover, it allows for easy generalization to other groups where such a simple identification is not possible (Steinberg, 1961).

## A.6 Equivariance of G-cluster expansion

The equivariance of the G-cluster expansion is easily verified by applying a transformation $g$ to the input,

$$\boldsymbol{B}^K_{\boldsymbol{\alpha}} \circ g = \sum_{\boldsymbol{k}} \mathcal{C}^{\boldsymbol{\alpha},K}_{\boldsymbol{k}} \boldsymbol{A}_{\boldsymbol{k}} \circ g$$

$$= \sum_{\boldsymbol{k}} \mathcal{C}^{\boldsymbol{\alpha},K}_{\boldsymbol{k}} \left( \sum_{\boldsymbol{k}'} \prod_t \rho_{k_t, k_t'}(g) \boldsymbol{A}_{\boldsymbol{k}'} \right)$$

$$= \sum_{\boldsymbol{k}'} \left( \sum_{\boldsymbol{k}} \mathcal{C}^{\boldsymbol{\alpha},K}_{\boldsymbol{k}} \prod_t \rho_{k_t, k_t'}(g) \right) \boldsymbol{A}_{\boldsymbol{k}'} \tag{42}$$

$$= \sum_{\boldsymbol{k}'} \left( \sum_{K'} \rho_{KK'}(g) \mathcal{C}^{\boldsymbol{\alpha},K'}_{\boldsymbol{k}'} \right) \boldsymbol{A}_{\boldsymbol{k}'}$$

$$= \sum_{K'} \rho_{KK'}(g) \boldsymbol{B}^{K'}_{\boldsymbol{\alpha}}.$$

### A.7 Completeness of the basis and Universality of G-MACE

We explain in which sense the basis $\boldsymbol{B}_{\boldsymbol{\alpha}}^K$ is a complete basis, and briefly sketch how to prove this claim. The argument is contained almost entirely in (Dusson *et al.*, 2022) and only requires a single modification, namely Step 3 below, using a classical argument from representation theory. We will therefore give only a very brief summary and explain that necessary change.

We start with an arbitrary equivariant property $\Phi^V$ embedded in $V$ where we have a representation, i.e. the actual target property is $\Phi$ is then given as a linear mapping from $V$ to $Z$. For technical reasons, we require that only finitely many entries $\Phi_K^V$ may be non-zero, but this is consistent with common usage. For example, if $G = O(3)$ and if $\Phi$ is a scalar, then $\Phi_0^V = \Phi$, while all other $\Phi_{LM}^V \equiv 0$. If $\Phi$ is a covariant vector, then $\Phi_{LM}^V$ is non-zero if and only if $L = 1$; and so forth. For other groups, the labeling may differ but the principle remains the same.

*1. Convergence of the cluster expansion.* The first step in our parameterisation is to approximate $\Phi^V$ in terms of a truncated many-body expansion (4). It is highly application-dependent on how fast this expansion converges. Rigorous results in this direction in the context of learning interatomic potentials can be found in (Bachmayr *et al.*, 2021; Thomas *et al.*, 2022). A generic statement can be made if the number of input particles is limited by an upper bound, in which case the expansion becomes exact for a finite $N$. This case leads to the uniform density result stated in Theorem 4.1. We adopt this setting for the time being and return to the pointwise convergence setting below.

In the uniform convergence setting we also require that the domain $\Omega$ is compact.

Throughout the remainder of this section we may therefore assume that an $N$ can be chosen as well as smooth components $\varphi^{(n)}$ such that the resulting model $\Phi^{V,N}$ approximates $\Phi^V$ to within a target accuracy $\epsilon$,

$$|\Phi_K^{V,N}(\boldsymbol{x}) - \Phi_K^V(\boldsymbol{x})| \leq \epsilon \qquad \forall \boldsymbol{x} \in \mathrm{msets}(\Omega).$$

*2. The density of the embedding.* As already stated in the main text, if the components $\varphi_K^{(n)}$ are smooth, and the embedding $\{\phi_k\}_k$ is dense in the space of one-particle functions (5) then it follows that the $\varphi_K^{(n)}$ can be expanded in terms of the tensor product basis $\phi_{\boldsymbol{k}} := \otimes_{s=1}^n \phi_{k_s}$ to within arbitrary accuracy. The precise statement is the following standard result of approximation theory: if $\mathrm{span}\{\phi_k\}_k$ are dense in $C(\Omega)$, then $\mathrm{span}\{\phi_{\boldsymbol{k}}\}_{\boldsymbol{k}}$ are dense in $C(\Omega^n)$. That is, for any $\epsilon > 0$, there exist approximants $p_K^{(n)}$ such that

$$\|\varphi_K^{(n)} - p_K^{(n)}\|_\infty \leq \epsilon.$$

*3. The density of the symmetrized basis.* The next and crucial step is to show that, if the $\varphi_K^{(n)}$ are equivariant, then the $p_K^{(n)}$ may be chosen equivariant as well without loss of accuracy. If the group $G$ is compact then the representations $\rho$ can be chosen unitary (Broecker, 1985). In that case, the argument from (Dusson *et al.*, 2022) can be used almost verbatim: let

$$\bar{p}^{(n)}(\boldsymbol{x}) := \int_G \rho(g)^{-1} p^{(n)}(g\boldsymbol{x}) \, H(dg),$$

where $H$ is the normalized Haar measure then $\bar{p}^{(n)}$ is equivariant by construction and

$$
\begin{aligned}
&\left|\varphi^{(n)}(\boldsymbol{x}) - \bar{p}^{(n)}(\boldsymbol{x})\right| \\
&= \left| \int_G \rho(g)^{-1} \Big( \varphi^{(n)}(g\boldsymbol{x}) - p^{(n)}(g\boldsymbol{x}) \Big) \, H(dg) \right| \\
&\leq \int_G \left| \varphi^{(n)}(g\boldsymbol{x}) - p^{(n)}(g\boldsymbol{x}) \right| H(dg) \\
&\leq \int_G \|\varphi^{(n)} - p^{(n)}\|_\infty \, H(dg) \leq \epsilon.
\end{aligned}
$$

If the group is not compact, then one can apply "Weyl's Unitary Trick" (see (Bourbaki, 1989), Ch. 3): first, one complexifies the group (if it is real) and then constructs a maximal compact subgroup $K_{\mathbb{C}}$ of the complexification. This new group $K$ will have the same representation as $G$ and because it is compact, that representation may again be chosen as unitary. Therefore, symmetrizing $p^{(n)}$ with

respect to $K_{\mathbb{C}}$ results in an approximant that is not only equivariant w.r.t. $K_{\mathbb{C}}$ but also equivariant w.r.t. $G$.

*4. The density of the basis $\boldsymbol{B}_\alpha^K$.* As the last step, one can readily observe that the symmetrization and cluster expansion steps can be exchanged. I.e. first symmetrizing and then employing the steps (7) result in the same model. Letting $\epsilon \to 0$ in the foregoing argument while fixing the number of particles $\#\boldsymbol{x}$ results in all errors vanishing. Note that this will in particular require taking $N \to \infty$.

*5. Pointwise convergence.* To obtain density in the sense of pointwise convergence we first introduce the *canonical cluster expansion* without self-interacting terms

$$\Phi_K(\boldsymbol{x}) = \sum_{n=0}^{\infty} \sum_{j_1 < \cdots < j_n} v_K^{(n)}(x_{j_1}, \ldots x_{j_n}).$$

The difference here is that the summation is only over genuine sub-clusters. Because of this restriction, the series is finite for all multi-set inputs $\boldsymbol{x}$. In other words, it converges in the pointwise sense.

One can easily see that $v_n$ can be chosen (explicitly) to make this expansion exact. After truncating the expansion at finite $n \leq N$ and then expanding the potentials $v_K^{(n)}$ one can exactly transform the canonical cluster expansion into the self-interacting cluster expansion. This procedure is detailed in (Dusson *et al.*, 2022; Drautz, 2020).

The arguments up to this point establish the claimed universality for the linear ACE model. The corresponding universality of the TRACE model follows immediately from (Darby *et al.*, 2022). Since a single layer of the MACE model is a TRACE model, this completes the proof of Theorem 4.1.

## A.8 Product of groups

Let $G_1$ and $G_2$ be two reductive Lie groups (or finite groups). Let $A_1$ and $A_2$ be the two structure constants of $G_1$ and $G_2$, $(d\rho_1, \rho_1)$ and $(d\rho_2, \rho_2)$ their continuous and discrete generators. One can define a representation of the direct product group $G_1 \times G_2$ as

$$\rho := \left( A_1 | A_2, n_1 n_2, \{d\rho_1(X_i) \otimes I_2 + I_1 \otimes d\rho_2(\tilde{X}_j)\}_{i,j}, \{\rho(h_1) \otimes I_2 + I_1 \otimes \rho(h_2)\}_{h_1,h_2 \in \mathbf{H}_1,\mathbf{H}_2} \right) \quad (43)$$

The following essential property holds: if $\rho_1$ and $\rho_2$ are irreducible representations of $G_1$ and $G_2$ then $\rho_1 \otimes \rho_2$ is an irreducible representation of $G_1 \times G_2$. Moreover, for reducible Lie groups, all the irreps of $G_1 \times G_2$ are of this form. Therefore, one can construct all the irreps of $G_1 \times G_2$ this way. It is of particular interest in the case of equivariant message passing networks on points clouds, where the group of interest is $G \times S_n$.

Let us give a non-trivial application of lie-nn for computing invariants for the product group of $O(3) \times S_3$. We consider a set of three vectors,

$$a = \begin{bmatrix} a_x \\ a_y \\ a_z \end{bmatrix} \quad b = \begin{bmatrix} a_x \\ a_y \\ a_z \end{bmatrix} \quad c = \begin{bmatrix} a_x \\ a_y \\ a_z \end{bmatrix}$$

A given vector belong to the product representation of the $l = 1$ vector of the $O(3)$ group and the natural representation of $S_3$. In lie-nn, we can define the product representation as follows,

```
rep = lie.group_product(lie.finite.Sn_natural(3), lie.irreps.O3(l=1, p=-1))
```

If one wants to know the unique permutation invariant vector that one can construct from the three vectors $a$, $b$ and $c$, one can use the following code,

```
qs = lie.infer_change_of_basis(
    lie.group_product(lie.finite.Sn_trivial(3), lie.irreps.O3(l=1, p=-1)),
    rep,
)
```

As expected, the output is the sum of the three vectors, $a+b+c$, corresponding to the only permutation invariant vector that can be constructed from the three vectors.

Consider now the case where we want to extract a permutation invariant scalar from the set of products of two vectors. In this case, we can use the following code,

```
qs = lie.infer_change_of_basis(
    lie.group_product(lie.finite.Sn_trivial(3), lie.irreps.O3(l=0, p=1)),
    lie.tensor_product(rep, rep),
)
```

In this code, we first construct the tensor product representation and then seek the permutation invariant scalar in it. The result is two distinct scalars,

$$e_1 = a_x^2 + a_y^2 + a_z^2 + b_x^2 + b_y^2 + b_z^2 + c_x^2 + c_y^2 + c_z^2$$

$$e_2 = a_x b_x + a_x c_x + a_y b_y + a_y c_y + a_z b_z + a_z c_z + b_x c_x + b_y c_y + b_z c_z$$

Finally we consider a case that would be very hard to do by hand. It is the case of extracting a permutation invariant $l = 2$ vector for this tensor product of the original set of three vectors.

```
qs = lie.infer_change_of_basis(
    lie.group_product(lie.finite.Sn_trivial(3), lie.irreps.O3(l=2, p=1)),
    lie.tensor_product(rep, rep),
)
```

The computation tells us that there exists two such vectors,

$$v_1 = \begin{bmatrix} -\sqrt{2}a_x a_z - \sqrt{2}b_x b_z - \sqrt{2}c_x c_z \\ -\sqrt{2}a_x a_y - \sqrt{2}b_x b_y - \sqrt{2}c_x c_y \\ \frac{\sqrt{6}a_x^2}{6} - \frac{\sqrt{6}a_y^2}{3} + \frac{\sqrt{6}a_z^2}{6} + \frac{\sqrt{6}b_x^2}{6} - \frac{\sqrt{6}b_y^2}{3} + \frac{\sqrt{6}b_z^2}{6} + \frac{\sqrt{6}c_x^2}{6} - \frac{\sqrt{6}c_y^2}{3} + \frac{\sqrt{6}c_z^2}{6} \\ -\sqrt{2}a_y a_z - \sqrt{2}b_y b_z - \sqrt{2}c_y c_z \\ \frac{\sqrt{2}a_x^2}{2} - \frac{\sqrt{2}a_z^2}{2} + \frac{\sqrt{2}b_x^2}{2} - \frac{\sqrt{2}b_z^2}{2} + \frac{\sqrt{2}c_x^2}{2} - \frac{\sqrt{2}c_z^2}{2} \end{bmatrix}$$

$$v_2 = \begin{bmatrix} -a_x b_z - a_x c_z - a_z b_x - a_z c_x - b_x c_z - b_z c_x \\ -a_x b_y - a_x c_y - a_y b_x - a_y c_x - b_x c_y - b_y c_x \\ \frac{\sqrt{3}a_x b_x}{3} + \frac{\sqrt{3}a_x c_x}{3} - \frac{2\sqrt{3}a_y b_y}{3} - \frac{2\sqrt{3}a_y c_y}{3} + \frac{\sqrt{3}a_z b_z}{3} + \frac{\sqrt{3}a_z c_z}{3} + \frac{\sqrt{3}b_x c_x}{3} - \frac{2\sqrt{3}b_y c_y}{3} + \frac{\sqrt{3}b_z c_z}{3} \\ -a_y b_z - a_y c_z - a_z b_y - a_z c_y - b_y c_z - b_z c_y \\ a_x b_x + a_x c_x - a_z b_z - a_z c_z + b_x c_x - b_z c_z \end{bmatrix}$$

### A.9 Symmetric Tensor products

The permutation group is an important concept in the context of tensor products. It can be useful to focus on a subset of the full tensor product space that exhibits certain permutation equivariance. For example, the spherical harmonics are defined as the permutation-invariant part of a tensor product.

The symmetric tensor product can be thought of as a change of basis, or projector, from the tensor product to the symmetric part of the tensor product. In the case of a tensor product of correlation order four we have,

$$S_\nu = B_{\nu;ijkl} x_i y_j z_k w_l \tag{44}$$

where $B$ is the change of basis that satisfies:

$$B_{\nu;ijkl} = B_{\nu;\sigma(ijkl)} \forall \sigma \in S_4 \tag{45}$$

We propose in `lie-nn` a new algorithm used to calculate $B$. The Symmetric Tensor Product is calculated using a tree structure, starting at the leaves and progressing towards the trunk. The leaves are the basis of the individual indices, and they are combined and constrained at each step to impose symmetry.

### A.10 Computing the irreps from input representations

For some groups, the computation of the generators $X$ can become a very involved task. However in most applications, the data itself is already given in a form of a representation. One approach proposed by (Finzi *et al.*, 2021) is to not work in the space of irreps but the space of polynomials of the input representation. This approach has the advantage of requiring little previous knowledge of

the group. However it is also much less efficient than using irreps. One alternative way is to consider polynomials of the input representation, that are reducible and then compute the block diagonalisation to project down to irreps subspace. One can then work directly as polynomials in this subspace and compute Clebsch–Gordan coefficients numerically. We provide routines in `lie-nn` to carry out these operations from any given input representation.

## A.11  Details of numerical experiments

### A.11.1  Jet Tagging

**Dataset**  The dataset (Butter *et al.*, 2019) was generated using a `Pythia`, `Delphes`, and `FastJet` (using cuts for the jet's kinematics on $\Delta\eta = 2$, $R = 0.8$) to simulate the response of the ATLAS detector at the Large Hadron Collider (LHC). The dataset is released under the "Creative Commons Attribution 4.0" license. The entire dataset contains 2 millions jets with a 60/20/20 for training, validation, and testing balanced splits.

**Model**  The model uses **3 layers** of the $G$-MACE architecture to generate the Lorentz group equivariant representation of each jet. For the 1 particle basis, we use a product of radial features on the Minkowski distances, and $SO(1,3)$ spherical harmonics. The radial features are computing by passing a logarithmic radial basis as in (Bogatskiy *et al.*, 2022) into a $[64, 64, 64, 512]$ MLP using SiLU nonlinearities on the outputs of the hidden layers. The internal representations used are $(0,0)$ and $(1,1)$. We use 72 channels for each representation. For the embedding, and readout out, we use similar achitectures to LorentzNet.

**Training**  Models were trained on an NVIDIA A100 GPU in single GPU training. Typical training time for the dataset is up to 72 hours. Models were trained with AMSGrad variant of Adam, with default parameters of $\beta_1 = 0.9$, $\beta_2 = 0.999$, and $\epsilon = 10^{-8}$. We used a learning rate of 0.0035 and a batch size of 64. The model was trained for 80 epochs with 2 epochs of linear learning rate warmup and followed by a phase of cosine annealing LR scheduling.

### A.11.2  3D shape recognition

**Dataset**  ModelNet10 (Wu *et al.*, 2015) is a synthetic 3D object point clouds dataset containing 4,899 pre-aligned shapes from 10 categories. The dataset is split into 3,991 ($80\%$) shapes for training and 908 ($20\%$) shapes for testing. We were unable to find a license.

**Model**  The model uses a three-layer encoder architecture following the PointNet++ one. We use an encoder of the full point cloud into sub-point clouds of sizes $[1024, 256, 128]$. Each PointNet layer maps a point cloud of size $N^t$ to one of size $N^{t+1}$. We compute the node features as the sum of the PointNet output and the MACE output,

$$h^{(t+1)} = \text{PointNet}(xyz^{(t)}, h^{(t)}) + \text{MACE}(xyz^{(t)}, h^{(t)}) \tag{46}$$

**Training**  Models were trained on an NVIDIA A100 GPU in single GPU training. The typical training time for the dataset is up to 12 hours. Models were trained with the AMSGrad variant of Adam, with default parameters of $\beta_1 = 0.9$, $\beta_2 = 0.999$, and $\epsilon = 10^{-8}$.

## A.12  Limitations and Future Work

The spectrum of potential applications of the present method is very large. In this paper, we focus on a subset of applications that have known benchmarks and baselines. A broader range of groups is implemented in the lie-nn library. Future work should focus on applying this architecture to tasks with domain-specific knowledge.

### A.13 Computational cost

#### A.13.1 Clebsch Gordan generation

**Generation time**   The Clebsch Gordan generation time depends on the size of the representations. In Table 3, we plot the generation time as a function of the size of the representations for different groups. The generation time is found to range from a matter of milliseconds for smaller representations to a few seconds for larger ones. The generation of CGs constitutes a preprocessing step, separate from the model inference computations. Therfore, the results can be stored for subsequent use, ensuring that this phase does not affect the model's overall performance.

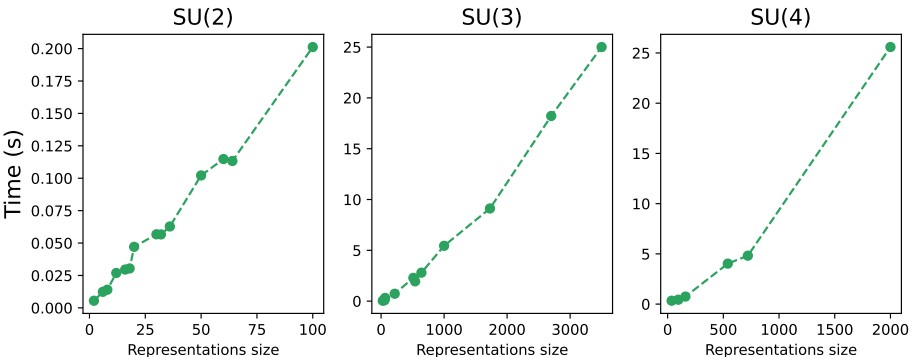

Figure 3: Generation time in seconds of Clebsch-Gordan coefficients as a function of the size of the representations for different groups. The representation size is computed as the product of the three representation sizes involved in the Clebsch-Gordan.

**Sparsity**   One distinguishing feature of the Clebsch-Gordan coefficient is their sparsity. This sparsity comes from the selection rule, which is unique for each group. These selection rules also give rise to a sparsity structure that can be exploited to achieve the best efficiency. In Figure 4 we compare the sparsity percentage as a function of the size of the representations for different groups.

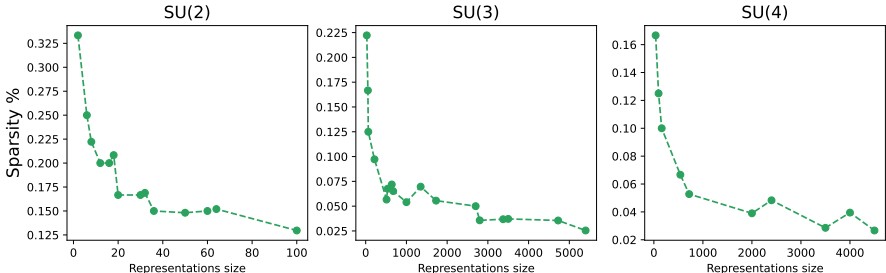

Figure 4: Sparsity in the percentage of non-zero entries of Clebsch-Gordan coefficients as a function of the size of the representations for different groups. The representations size is computed as the product of the three representation sizes involved in the Clebsch-Gordan.

#### A.13.2 G-MACE computational cost

The computational bottleneck of traditional equivariant MPNNs is the equivariant tensor product on the edges of the graph used to form the message. In G-MACE the edge operation is the pooling operation to compute the features $A_k$. Correlations $\boldsymbol{A_k}$ are computed through the tensor product of the product basis, an operation that is carried out on nodes. In typical models, the correlations are the bottleneck. Since the number of nodes is orders of magnitudes smaller than the number of edges in most applications we are envisioning, it is a significant computational advantage to organize the computation in this way. We return to reviewing the cost of the product basis below.

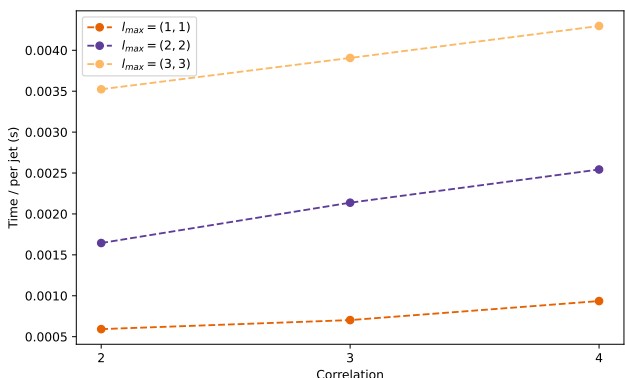

Figure 5: Inference time for a single jet for a two layer LorentzMACE model as a function of the correlation order in the product basis and the maximal angular resolution. Timing made on a Nvidia A100 and averaged over a 100 runs.

Obtaining the equivariant basis $\boldsymbol{B}^K$ involves simply linear operation. The Clebsch-Gordan coefficients are very sparse, with a well defined structural sparsity, which can be easily exploited for constructing highly efficient low level code on both CPUs and GPUs.

Thus, we return to the product basis $\boldsymbol{A}_{\boldsymbol{k}}$, which is normally the computational bottleneck. It has structural sparsity due to its symmetry under permuting the $\boldsymbol{k}$ tuples and due to the structural sparsity in the Clebsch–Gordan coefficients. Despite that sparsity it was shown in Kaliuzhnyi and Ortner (2022) that *in theory* the product basis can be evaluated recursively with $O(1)$ operations per feature, however, this requires effective use of sparse tensor formats which is challenging to optimize for GPU architectures since a naive implemention relies on random memory access patterns. Instead, the *current* MACE code employs a similar recursive evaluation algorithm described in Batatia *et al.* (2022a), which employs full tensor formats, but avoids the random memory access. For the most common case of 3-correlation models we estimate that this code reaches within a factor 3-5 of the optimal performance of hypothetical sparse tensor implementation.

