# OpenReview forum: "A General Framework for Equivariant Neural Networks on Reductive Lie Groups"
_NeurIPS.cc/2023/Conference — NeurIPS 2023 poster_

### Official Review · Reviewer_KJSX · 2023-07-02

**Soundness:** 3 good
**Presentation:** 3 good
**Contribution:** 3 good
**Rating:** 5
**Confidence:** 5

**Summary:**

The paper presents a novel and highly general Equivariant Neural Network (ENN) architecture that is capable of respecting the symmetries of the finite-dimensional representations of any reductive Lie Group G. The proposed approach generalizes the successful ACE and MACE architectures for atomistic point clouds to any data equivariant to a reductive Lie group action. The authors demonstrate the generality and performance of their approach by applying it to two different tasks, namely top quark decay tagging (Lorentz group) and shape recognition (orthogonal group). The results presented in the paper are convincing and showcase the potential of the proposed architecture.

**Strengths:**

* The paper is well-organized and well-written, providing a clear overview of the problem and proposed solution.
* The paper addresses a significant and challenging problem by proposing a highly general ENN architecture that can handle symmetries of the finite-dimensional representations of any reductive Lie Group G, which contributes to the field of equivariant neural networks.

**Weaknesses:**

Related concerns are discussed in the questions section.

**Questions:**

* A more comprehensive discussion comparing methods based on Lie Groups, such as LieConv, should be included in the paper.
* The results presented in the paper might not adequately convey the effectiveness of the proposed method. In order to reinforce the claims and emphasize the practical applicability of the approach, it would be valuable for the authors to incorporate comparisons of experimental results from methods like MACE and LieConv, using examples such as those from the QM9 dataset.

**Limitations:**

There are no potential negative societal impacts of this work.

---

> ### Author Rebuttal · Authors · 2023-08-09
>
> Thank you for your time and effort in reviewing our paper. We appreciate
> that you find our paper well-organized and well-written and addresses a
> significant and challenging problem. You will find our response to your
> remarks and questions here. We respectfully hope that our responses will
> be satisfactory and will increase your mark.
>
> ## More in depth discussion of LieConv
>
> > A more comprehensive discussion comparing methods based on Lie Groups,
> > such as LieConv, should be included in the paper.
>
> In response to your comment, we have incorporated a new section in the
> appendix, entitled \"Extended Related Work.\" This section delves into a
> detailed examination of previous literature, including the LieConv
> method. It is important to note that our work stands out in providing a
> more comprehensive framework than any previously developed architecture,
> both in terms of the diversity of groups covered and the level of
> expressiveness achieved. To our knowledge, our method represents the
> first instance of a higher-order equivariant neural network for generic
> reductive Lie groups.
>
> Moreover, the applicability of LieConv is confined to compact groups, as
> it necessitates the explicit computation of an integral over the group.
> The convolution in LieConv is limited to considering two-body
> interactions, whereas our approach, G-MACE, accommodates interactions of
> arbitrary order. To this day and for the reasons outline above, LieConv
> is only implemented for a restricted number of groups namely abelian groups and subgroups of
> the $O(3)$ group. Our method and library covers a much broader range of
> applications in physics and beyond. To the best of our knowledge, no
> previous work has provided such a general framework.
>
> Convolutional Neural Networks (CNNs), which are translation equivariance, initiated the utilization of data symmetry in machine learning architectures. Throughout time, CNNs have been extended to include other symmetries as well. Central to all these generalizations (including LieConv [2]) is the group averaging operation,
>
> $$
>     \text{Avg}(f)(x) = \int_{g \in G} f(g \cdot x) dg,
> $$
>
> Where $ x $ denotes the input signal or feature, $f$ is the convolution kernel, $G$ represents the group of interest, and $dg$ is an invariant measure on $G$. This transformation is essential, as it converts any convolution into a group invariant convolution. The feasibility of this approach largely depends on the computational simplicity of the integral. This approach has several limitations,
>
> - The direct computation of the integral is unstable and inefficient, even for relatively small groups like $O(3)$.
> - For non-compact groups, a unique invariant measure is absent, and the integral diverges.
> - The convolution kernel $f$ is usually constrained to a two-body operator .
>
> In the case of compact groups, the integral over the group may be calculated via an alternative means. There exists a linear operator, called the Clebsch Gordan operator $\mathcal{C}$, such that,
> \begin{equation}
>      \text{Avg}(f)(x) = \mathcal{C}(f)(x)
> \end{equation}
> Therefore, the complex integral over the group becomes a linear operation. The central aim of our work is to show that this approach can also be generalized to all reductive Lie groups, even non-compact ones, and provide tools to do so including the basis to expand $f$ and tools to compute $\mathcal{C}$ in this basis.
>
> ## More comparaison to LieConv
>
> > In order to reinforce the claims and emphasize the practical
> > applicability of the approach, it would be valuable for the authors to
> > incorporate comparisons of experimental results from methods like MACE
> > and LieConv, using examples such as those from the QM9 dataset.
>
> MACE is a subset of the presented G-MACE architecture, when the group is
> the rotation group and the task is on molecular data. Independently to
> this work, the MACE force field architecture has been recently
> benchmarked on QM9 and compared to LieConv in \[1\]. Here are selected
> comparisons relevant to this paper,
>
> |                      | $\textbf{Gap}$ | $\textbf{Homo}$ | $\textbf{Lumo}$ | $C_{V}$ | $\mu$ | $\textbf{ZPVE}$ |              |
> |----------------------|--------------|---------------|---------------|--------------|------------|---------------|--------------|
> |                      | meV          | meV           | meV           | cal/mol K    | D          | meV           | $\alpha_0^2$ | $\alpha_0^3$ | meV | meV | meV | meV |
> | $\textbf{LieConv}$ [2] | 49           | 30            | 25            | 0.038        | 0.032      | 2.28          | 0.800        | 0.084        | 22  | 24  | 19  | 19  |
> | $\textbf{MACE}$ [1]    | 42           | 22            | 19            | 0.021        | 0.015      | 1.23          | 0.210        | 0.038        | 5.5 | 4.7 | 4.1 | 4.1 |
>
> From this table, you can observe that MACE far outperforms LieConv,
> thanks to more expressiveness due to its higher-order interactions. In
> the paper, we compare G-MACE to an extensive range of methods for the
> Lorentz group and have now included more baselines for the point-cloud
> classification. In both of these tasks, G-MACE also achieves
> state-of-the-art performance. Beyond QM9, the other application
> benchmark of LieConv is the toy image dataset RotMNIST, corresponding to
> the group $SO(2)$. This group is implemented in our library. However, as
> it is both an abelian and compact group, specialized architecture has
> been constructed for this case that performs very well, including
> steerable convolution. In the paper, we have preferred to focus on
> problems in which the point cloud representation is more natural than
> for images and the group theory more challenging.
>
> \[1\] Evaluation of the MACE Force Field Architecture: from Medicinal
> Chemistry to Materials Science, D.P Kovacs, I. Batatia, E.S. Arany, G.
> Csanyi
>
> \[2\] Generalizing Convolutional Neural Networks for Equivariance to Lie
> Groups on Arbitrary Continuous Data, M. Finzi, S. Stanton, P. Izmailov,
> A. G. Wilson

---

> > ### Comment · Reviewer_KJSX · 2023-08-15
> > **Thanks for your response**
> >
> > I appreciate the additional details provided in the rebuttal, as they have addressed the majority of my questions and concerns. Therefore, I will increase the rating.

---

### Official Review · Reviewer_5uTH · 2023-07-06

**Soundness:** 3 good
**Presentation:** 3 good
**Contribution:** 3 good
**Rating:** 6
**Confidence:** 4

**Summary:**

The paper proposes a class of G-equivariant neural networks for reductive lie groups that generalizes the previous ACE and MCAE models (which are designed to be equivariant with respect to the orthogonal group O(3)) to more general irreducible Lie groups.  A software library (Lie-NN) has also been developed and released for implementation of the proposed method. The experimental results on jet tagging and 3d point cloud classification support the authors' claims about the equivariant model improving model performance.

**Strengths:**

- The proposed approach seems novel; I am not aware of other works that design a G-equivariant network in this way.
- The paper is generally well-written and technically correct (although I have not performed an in-depth check for the proof of the universality theorem). Occasional variable definitions are missing but I had no trouble following the general flow of the paper.
- The experimental results appear to validate the claimed advantages of the proposed method (although more discussion on the computational aspects is needed to assess practicality; see below).

**Weaknesses:**

- Since this work is a generalized version of the previous ACE and MACE methods, a brief review of ACE and MACE would be helpful to understand and evaluate the novelty of the method. The current version is presented in a way that contains elements of previous existing methods, without a clear delineation of the new and original aspects of the proposed method. Exactly what is the precise nature of the extensions to ACE and MACE should be made clearer.
- While the technical contents seem correct, some effort at providing intuitive explanation and justifications at key places would be helpful to understanding the paper.  Descriptive figures, intuitive examples (e.g., basis including 1-particle basis or Clebsch-Gordan coefficients) come to mind as examples. In particular, I am curious about how to construct a 1-particle basis for specific Lie groups, e.g., SO(3).
- The formulation in the case of the product group in Section 4 seems to be missing, although there is a mention of the product group in the introduction and in Section 5. Is it trivial to design an equivariant model for the product group G_1 x G_2 with the formulation used in this method?  My initial impression is that there may be some subtleties involved, such as the order of the group actions.
- A discussion of the computational aspects of this method is missing.  When using this model, the Clebsch-Gordan coefficients must be calculated numerically (as mentioned in Section 5), and it would be helful to mention calculation times and errors. Does the calculation time vary depending on the number of basis components? I am also curious about how much time it takes for iteration on backpropagation compared to classic MLP models.
- Continuing with the above comment, since the Clebsch-Gordan coefficients are numerically calculated, some numerical calculation errors are inevitable. Is the equivariance of the model maintained in the presence of numerical errors? If not, some discussion, even qualitative, should be provided (e.g., to what extent the model remains equivariant) although quantitative results (e.g., experimental results, figures) would clearly be preferable.


**Questions:**

These have been mentioned for the most part in the weaknesses section.

**Limitations:**

No potential negative societal impact of this work as far as I can tell. As mentioned earlier, computational aspects of the method would be helpful. I don't want the authors to feel compelled to show that this method is immediately computationally practical, as that is not the only measure of the value and worth of any new idea, but it would be helpful to indicate to other researchers on what the computational limitations are, and to possibly spur interest in finding improvements.

---

> ### Author Rebuttal · Authors · 2023-08-09
>
> We thank you for your time and effort in reviewing our paper. We
> sincerely appreciate your thoughtful review. We appreciate that you have
> found our paper to be novel and well-written. You will find here-after
> our responses to your questions and remarks, including the changes we
> made to the paper following your review.
>
> ## Clarification on extension of ACE and MACE
>
> > Exactly what is the
> > precise nature of the extensions to ACE and MACE should be made
> > clearer.
>
> There are two aspects to this question: (1) the general architectures;
> and (2) the generation of the generalized Clebsch-Gordan coefficients.
>
> \(1\) The architecture of the models we present here is a
> generalization of the special case of $O(3)$-equivariance treated in
> previous works. However, for most researchers in the field, it
> is unclear whether this extension is possible or how to formulate generalization. The
> purpose of Section 4 is to make precise that this generality is indeed
> possible and how it can be achieved. To the best of our knowledge this
> has not yet been communicated anywhere else.
>
> \(2\) To make the general framework from Section 4 practical, the most
> challenging aspect is generating the symmetrisation operation, i.e., the
> generalized Clebsch--Gordan coefficients. This is achieved in Section 5.
> This is technically challenging and novel work leading to new software
> that has a potentially widespread impact across many application areas, as
> indicated throughout the manuscript.
>
> In response to the question, we lightly edited the list of contributions
> at the end of Section 1, but in terms of writing style, we prefer to keep
> it a bit understated.
>
> ## Examples of 1-particle basis
>
> > Some effort at providing intuitive explanation and justifications at
> > key places would be helpful to understanding the paper. \[\...\] In
> > particular, I am curious about how to construct a 1-particle basis for
> > specific Lie groups, e.g., SO(3).
>
> Following your remark, we have added an extensive discussion of the
> 1-particle basis in the Appendix, giving examples for the $SO(3)$ group and the
> Lorentz group. We hope this will clarify your questions.
>
> ## On product of groups
>
> > Is it trivial to design an equivariant model for the product group
> > $G_1 \times G_2$ with the formulation used in this method? My initial
> > impression is that there may be some subtleties involved, such as the
> > order of the group actions.
>
> Our library, `lie-nn`, functions at the level of matrix representations,
> making the integration of product groups a natural step. We now provide
> a more comprehensive explanation of this in the Appendix (see Product of
> groups), where we include a practical example of calculating non-trivial
> invariants for the product groups $O(3) \times S_{n}$. It is worth
> noting that the direct product of groups is commutative. Specifically,
> for any two groups $G_{1}$ and $G_{2}$, we have the isomorphism
> $G_{1} \times G_{2} \cong G_{2} \times G_{1}$.
>
> ## Computational cost
>
> > A discussion of the computational aspects of this method is missing.
> > Does the calculation time vary depending on the number of basis
> > components?...
>
> We added a discussion to the Appendix of the paper and a brief sentence
> in Sec 4.1. to reference that discussion. In the Appendix, we make an
> analysis of the computational cost of G-MACE as a function of the
> correlation order and order of expansion in the 1-particle basis, in the
> case of the Lorentz group.
>
> In general, it depends on hyperparameter choices, but especially for
> larger models the product basis $A_{k}$ is the theoretical and
> practical bottleneck. *In theory* the product basis can be computed at
> O(1) cost per feature (see arXiv:2202.04140, in fact it is proven that
> the cost is asymptotically 2 operations per feature), but the current
> G-MACE implementation does not leverage this algorithm yet as it requires
> efficient use of sparse tensors, which is difficult on GPU architectures
> as a naive implementation would require random memory access. At present
> the G-MACE code uses a highly performant dense tensor format
> implementation which we believe achieves within a factor of 3-5 of the
> hypothetical, optimal performance of a sparse code in the low to moderate correlation order regime that is relevant in applications. The code published with this article generalizes
> this efficient GPU implementation used in $O(3)$-MACE to handle contractions
> in any groups.
>
> Backpropagation is of comparable cost to inference. The computational kernels we employ are relatively simple. The
> backward pass is 2-5 times more expansive than the forward pass.
>
> Classical MLP models exploit BLAS3 and similar dense tensor operations
> that have been optimized *ad nauseum* by generations of researchers. Our codes do not yet reach a similar level
> of peak FLOPs performance and hence the cost of our models will be larger *per parameter*. But note that our physical priors (such as symmetries) usually result in
> models that are much smaller and more data-efficient.
>
> ## Numerical error and generation of Clebsch Gordan
>
> > Since the Clebsch-Gordan coefficients are numerically calculated, some
> > numerical calculation errors are inevitable. It
> > would be helpful to mention calculation times and errors.
>
> While numerical errors are generally inevitable, we use numerically stable solvers
> for solving the linear systems, thereby achieving machine precision
> errors. Moreover, for $SU(N)$, the Clebsch-Gordan (CG)
> coefficients are roots of rational fractions. We employ a
> rounding scheme designed to round to the nearest rational fractions,
> reaching exact accuracy in this case.
>
> Following your remark regarding calculation times, we have
> added a section to
> the Appendix including a comparison of CG
> generation across various sizes of representations and different groups (see pdf in general response).
> We would like to underscore that the generation of CGs constitutes a
> preprocessing step; so, this phase does not affect the model's performance.

---

> > ### Comment · Reviewer_5uTH · 2023-08-15
> >
> > I appreciate the follow-up to my queries; my questions and concerns have been sufficiently addressed. I'm still left with the impression that computational and numerical considerations for this method are important and yet are not mentioned as prominently as they should be in the main body of the paper (only in the appendix). Perhaps there's not much the authors can do about this, since revisions to the main body of the paper are not allowed at this stage.

---

> > > ### Author Response · Authors · 2023-08-15
> > >
> > > We are glad to know that our response was satisfactory to you. We plan to include all the new experiments related to computational and numerical considerations in the main part of the paper. This will be done as soon as we are permitted an additional page for the final version. We hope this plan is satisfactory to you.

---

### Official Review · Reviewer_Mk5u · 2023-07-06

**Soundness:** 4 excellent
**Presentation:** 3 good
**Contribution:** 3 good
**Rating:** 8
**Confidence:** 4

**Summary:**

The authors generalize MACE, a point cloud network that uses higher-order interactions via tensor products of basis expansions of the features, to being equivariant to arbitrary reductive Lie groups. The paper shows that this setup inherits universality properties from MACE. A generic method to compute the a basis for the equivariant linear maps between tensor products of the representations is implemented.
While the paper is mostly theoretical, the authors show strong performance on a SO(1, 3) equivariant task and a point cloud task.

**Strengths:**

- The proposed method is an elegant generalization of a popular prior method
- The paper is well-written and mostly easy to read.
- The code is included
- The proposed method could be a useful turn-key equivariance solution for researchers with a niche symmetry problem.

**Weaknesses:**

- Figure 1 of the paper appears to suggest that the method works on E(3), but this can't be a reductive group, as the 4D homogenous representation is not decomposable as a sum of irreducibles (contradicting lines 67-68). I suppose the authors mean to write O(3) and treat the translation by canonicalization, but they should clarity that.
- The point could experiment should include more baselines.
- As the key contribution of the paper mostly lies in implementing the necessary computations for generic reductive Lie groups (sec 5), it would make sense to allocate more space to that and less to the prior sections, which are mostly already covered in prior works. Alternatively / in addition, it would be good to give more background on the material in section 5 in the appendix.

**Questions:**

- It would be helpful to include/repeat the definition of B used in equation (20).

**Limitations:**

The authors should clarify better that not all Lie groups of interest are reductive.

---

> ### Author Rebuttal · Authors · 2023-08-09
>
> Thank you very much for your positive review and excellent comments. We
> appreciate that you find our work interesting and well written. Below we
> respond to your questions and suggestions.
>
> ## Clarification on $E(3)$
>
> > Figure 1 of the paper appears to suggest that the method works on
> > E(3), but this can't be a reductive group, as the 4D homogenous
> > representation is not decomposable as a sum of irreducibles
> > (contradicting lines 67-68). I suppose the authors mean to write O(3)
> > and treat the translation by canonicalization, but they should clarity
> > that.
>
> We thank you for spotting this typo. As you said, the translation
> group is not reductive, and we usually use canonicalization to
> construct invariants. Now we refer to the $O(3)$ group in the figure.
> One interesting aspect of the translation group is that it is an abelian
> group. Therefore the irreducible representations are still well
> understood, and our framework could extend to it. We intend to include
> representations of such groups in further work.
>
> ## Baseline on point clould experiment
>
> > Point could experiment should include more baselines.
>
> We now compare to other state-of-the-art models for 3D shape recognition
> in Table 3. We have selected the best models we could find that use
> point cloud or voxel representations. If we are missing any, we would be
> happy to reference it and, if appropriate, add it to the benchmark. Please find here a
> copy of this updated table,
>
> | Architecture |  **PointMACE** (ours) | **PointNet**  |  **PointNet ++**    | **KCN**  | **SO-Net** | **LP-3DCNN**|
> | --- | --- | --- | --- | --- | --- | --- |
> | Accuracy | **96.1** | 94.2 |  95.0 | 94.4 | 95.5 | 94.4 |
> | Representation | Point Cloud | Point cloud | Point cloud | Point Cloud | Point Cloud | Voxel grid |
>
>
> Note that we compare only to other point cloud methods. The best current
> model we are aware of uses additional information in images at different
> angles and achieves an accuracy of about 98 %. We believe that this
> omission is fair.
>
> ## Balance in the sections
>
> > As the key contribution of the paper mostly lies in implementing the
> > necessary computations for generic reductive Lie groups (sec 5), it
> > would make sense to allocate more space to that and less to the prior
> > sections
>
> Following your remark, we have shortened the first sections and added a
> new subsection to the main text discussing the symmetric powers of
> representations of reductive Lie groups and how to generate the
> generalized Clebsch Gordan. Moreover, we have added an extensive (5 pages)
> discussion of the background of reductive Lie groups, Lie algebras, and
> GT patterns. Please find that in the
> new Appendix section A.1. We want to emphasize that we tried to put the
> most technicalities in the appendix, as we want this paper to be
> accessible to a broad audience. For such an audience, we think that a
> certain detail of the general $G$-equivariant cluster expansion
> techniques is important. In particular, it makes precise how the ideas
> used in the $O(3)$ case generalize.

---

> > ### Comment · Reviewer_Mk5u · 2023-08-11
> > **Thanks for the response**
> >
> > I thank the authors for their response. My score remains unchanged.

---

### Official Review · Reviewer_jt5w · 2023-07-07

**Soundness:** 3 good
**Presentation:** 3 good
**Contribution:** 3 good
**Rating:** 6
**Confidence:** 2

**Summary:**

This paper proposes a framework for building equivariant neural networks on reductive lie groups. The proposed method first constructs a linear model for multi-set functions which is then symmetrised to generate a complete basis of equivariant multi-set functions. The model is also extended to a multi-layer architecture by a message-passing scheme.

**Strengths:**

* The idea of generalizing ACE and MACE frameworks to arbitrary reductive Lie groups is new.
* Proofs are given in the supplemental material to support theoretical claims in the paper.
* A library is provided for developing G-equivariant neural networks.


**Weaknesses:**

* The paper has some minor errors in writting. For instance, Table 2 caption should be on top of the table. Tables (e.g., Tabs. (2) and (3)) that report experimental results should be mentioned in the text.
* Lack of comparison against state-of-the-art methods for validating the effectiveness of the proposed method on 3D shape  recognition.


**Questions:**

* The computations in Eqs. (7) together with (16) seem to be expensive. How these can be done in practice ?
* It would be interesting to show the impact of higher order messages on the performance of G-MACE in terms of accuracy and computation time. I didn't find this study in the paper and supplemental material.

**Limitations:**

Limitations are discussed in the supplemental material.

---

> ### Author Rebuttal · Authors · 2023-08-09
>
> Thank you for reviewing our paper. We appreciate that you find our work
> novel and our open-source library interesting to the community.
> Below we respond to your questions and suggestions to further improve
> the paper.
>
> ## Formatting of tables
>
> > Table 2 caption should be on top of the table. Tables (e.g., Tabs. (2)
> > and (3)) that report experimental results should be mentioned in the
> > text.
>
> We thank you for spotting these problems. We have fixed the caption and
> we are now cross-referencing the tables in the text.
>
> ## Baseline for 3D shape recognition
>
> > Lack of comparison against state-of-the-art methods for validating the
> > effectiveness of the proposed method on 3D shape recognition.
>
> We now compare to other state-of-the-art models for 3D shape recognition
> in Table 3. We have selected the best models we could find that use
> point cloud or voxel representations. If we are missing any, we would be
> happy to reference it and, if appropriate, add it to the benchmark. Here is a copy of this updated table,
>
> | Architecture |  **PointMACE** (ours) | **PointNet**  |  **PointNet ++**    | **KCN**  | **SO-Net** | **LP-3DCNN**|
> | --- | --- | --- | --- | --- | --- | --- |
> | Accuracy | **96.1** | 94.2 |  95.0 | 94.4 | 95.5 | 94.4 |
> | Representation | Point Cloud | Point cloud | Point cloud | Point Cloud | Point Cloud | Voxel grid |
>
> Note that we compare only to other point cloud methods. The best current
> model we know of uses additional information in images at different
> angles, achieving an accuracy of about $98$ %. We believe that this
> omission is fair.
>
> ## Efficient implementation of Equivariant Product Basis
>
> > The computations in Eqs. (7) together with (16) seem to be expensive.
> > How these can be done in practice ?
>
> In general, it depends on hyperparameter choices, but especially for
> larger models the product basis $\textbf{A}_{\textbf{k}}$ is indeed the
> theoretical and practical bottleneck. *In theory* the product basis can
> be computed at O(1) cost per feature (see arXiv:2202.04140, in fact it
> is proven that the cost is asymptotically two operations per feature), but
> the current G-MACE implementation does not leverage this algorithm yet as
> it requires efficient use of sparse tensors, which is difficult on GPU
> architectures as a naive implementation would require random memory
> access.
>
> At present, the G-MACE code uses a highly performant dense tensor format
> implementation, which we believe achieves within a factor of 3-5 of the
> hypothetical, optimal performance of a sparse code, in the low to moderate correlation order regime that is relevant in applications.
> This renders its cost very attractive. The code published with this article generalizes
> this efficient implementation used in $O(3)$-MACE to handle contractions
> in any group.
>
> We added a discussion to the appendix of the paper and a brief sentence
> in Sec 4.1. to reference that discussion.
>
> ## Impact of higher order in accuracy and speed
>
> > The impact of higher order messages on the performance of G-MACE in
> > terms of accuracy and computation time.
>
> We refer to the previous question: a theoretical optimal evaluation
> scheme requires only O(1) operations per feature, independent
> of the correlation order. In practice, the current implementation's cost depends on correlation order. Since all our models
> seem to be optimal in the range of correlation order 2, 3, 4 (typically
> 3) and since we are actively working towards the implementation of a
> a quasi-optimal algorithm, we prefer not to emphasize this too much in the
> paper. In the Appendix A.13.2, we have added an analysis of the computational cost of
> G-MACE as a function of the correlation order in the case of the Lorentz
> group.  Please find the figures of this section in the pdf attached to the general response.
>
> While in theory, higher correlation means more expressiveness, the relationship between accuracy and the correlation order depends on the dataset. Please find here a table summarizing the accuracy and computational time for different correlation orders on the jet tagging dataset,
>
> |  **Correlation** | 1 | 2 |  3    |  4|
> | --- | --- | --- | --- | --- |
> | **Accuracy** | 93.6 | **94.2** |  **94.2** | **94.2** |
> | **Timings** (ms / jet) | **0.35** | 0.58 | 0.71 | 0.93|
>
> In the case of the jet dataset, we see significant improvement going from correlation order one to two and then saturation. In molecular applications, correlation order three is routinely used. It is essential to note that two layers of correlation order three at each layer, give rise to functions of correlation 12. The convergence of the many-body expansion highly depends on data. Low body order is likely enough if the physics behind the data is close to a mean-field limit.

---

> > ### Comment · Reviewer_jt5w · 2023-08-21
> >
> > I thank the authors for their adequate answers. I maintain my original rating.

---

### Author Rebuttal · Authors · 2023-08-09

We thank all reviewers for their time and effort in reviewing our paper.
We are glad you think that our work is "new" (R1) and that our "proposed method is an elegant generalization of previous methods" (R2), addressing "a significant and challenging problem" (R4) of the field of equivariant neural networks. We appreciate that you find our manuscript "well-written" and "well-organized" (R3).

In summary, we have updated our manuscript with the following changes. An updated manuscript incorporating those changes will be made available on arxiv within a few days of the deadline.

-  We have improved the overall clarity of the writing and referencing of results.
-  We have added a new subsection to section 5 in the main text on symmetric powers of representations and the generation of generalized Clebsch Gordan coefficients for reductive Lie groups.
-  Fixed the typo in Figure 1.
-  We added a more extensive number of baselines for the 3D shape recognition task. Here you will find the updated table:
| Architecture |  **PointMACE** (ours) | **PointNet**  |  **PointNet ++**    | **KCN**  | **SO-Net** | **LP-3DCNN**|
| --- | --- | --- | --- | --- | --- | --- |
| Accuracy | **96.1** | 94.2 |  95.0 | 94.4 | 95.5 | 94.4 |
| Representation | Point Cloud | Point cloud | Point cloud | Point Cloud | Point Cloud | Voxel grid |

- We added an extensive background on Lie groups and their representation in the Appendix. We expose in more detail the theoretical foundations of our work. In particular, we give a more in-depth summary of the Gelfand-Tsetlin patterns.

- We added a new section to the Appendix that gives concrete examples of one particle basis for the case of the $O(3)$ group and the Lorentz group.

- We give more details for constructing irreducible representations of the product of groups. We also provide a concrete application of the lie-nn library for computing invariants of the product groups $O(3) \times S_{3}$.

- We have added a section in the Appendix on computational cost, both on Clebsch Gordan generation for a wide range of groups and comparing the cost of ablated versions of G-MACE for the Lorentz group. Please find the figures in the pdf attached.

- We have incorporated an Extended Related Work section to the Appendix, providing an extended background on methods for constructing equivariant neural networks. This section also helps to clarify our contributions.

- Finally, we have published our code on GitHub for complete reproducibility.

Below we respond to the reviewers' comments individually.

---

> ### Comment · Reviewer_Mk5u · 2023-08-11
> **Appropriate to refer to arxiv?**
>
> Dear authors and AC,
>
> Given that the reviewers are not allowed to seek out this paper on arxiv and this conference does not permit uploading a revised version of the paper, I believe it is inappropriate to refer to arxiv, or to any existing updated version of the paper in general. Perhaps the authors can remove the reference to arxiv in their rebuttal and rephrase their rebuttals from "we have changed/fixed" to "we will change/fix".
>
> Thanks

---

> > ### Author Response · Authors · 2023-08-11
> >
> > Dear Reviewers and AC,
> >
> > We now realize that this was indeed inappropriate and put this mistake on our haste and confusion on the last minutes before the deadline.
> > For future readers, please ignore this line in the rebuttal.
> >
> > If the AC could allow us to remove that line from the rebuttal as we can not do it now?

---

> > > ### Comment · Area_Chair_3D2Y · 2023-08-15
> > > **ArXiv**
> > >
> > > I agree that it would have been better not to refer to the arxiv preprint, since reviewers are expected not to look at the de-anonimized arxiv paper. I cannot remove the line in the review, but I expect that the reviewers will be able to disregard it.

---

### Decision · Program_Chairs · 2023-09-21

**Decision:**

Accept (poster)

**Comment:**

The paper introduces a general method for construction of equivariant networks for arbitrary reductive Lie groups, generalizing the ACE / MACE methods. Reviewers found the work novel, clearly written, and elegant, and appreciated the release of code. The main concerns raised by the reviewers were addressed in the rebuttal/discussion, and so I recommend this paper to be accepted.